# Meta Reinforcement Learning with Finite Training Tasks - a Density Estimation Approach

**Zohar Rimon**
Technion - Israel Institute of Technology
zohar.rimon@campus.technion.ac.il

**Aviv Tamar**
Technion - Israel Institute of Technology
avivt@technion.ac.il

**Gilad Adler**
Ford Research Center Israel
gadler3@ford.com

## Abstract

In meta reinforcement learning (meta RL), an agent learns from a set of training tasks how to quickly solve a new task, drawn from the same task distribution. The optimal meta RL policy, a.k.a. the Bayes-optimal behavior, is well defined, and guarantees optimal reward *in expectation*, taken with respect to the task distribution. The question we explore in this work is *how many* training tasks are required to guarantee approximately optimal behavior with high probability. Recent work provided the first such PAC analysis for a model-free setting, where a history-dependent policy was learned from the training tasks. In this work, we propose a different approach: directly learn the task distribution, using density estimation techniques, and then train a policy on the learned task distribution. We show that our approach leads to bounds that depend on the dimension of the task distribution. In particular, in settings where the task distribution lies in a low-dimensional manifold, we extend our analysis to use dimensionality reduction techniques and account for such structure, obtaining significantly better bounds than previous work, which strictly depend on the number of states and actions. The key of our approach is the regularization implied by the kernel density estimation method. We further demonstrate that this regularization is useful in practice, when 'plugged in' the state-of-the-art VariBAD meta RL algorithm.

## 1 Introduction

In recent years, reinforcement learning (RL) became a dominant algorithmic framework for a variety of domains, including computer games [24], robotic manipulation [11], and autonomous driving [18]. Popular RL algorithms, however, are characterized by a high sample inefficiency, due to the exploration-exploitation problem – the need to balance between obtaining more information about the environment versus acting based on such information. Indeed, most RL success stories required a very long training process, only possible in simulation.

For an agent to learn fast, additional structure of the problem is required. In Meta RL [4; 39; 7; 26], agents are allowed to train on a set of training tasks, sampled from the same task distribution as the task they will eventually be tested on. The hope is that similar structure between the tasks could be identified during learning, and exploited to quickly solve the test task. It has recently been observed that the meta RL problem is related to the Bayesian RL formulation, where each task is modelled as a Markov decision process (MDP), and the distribution over tasks is the *Bayesian prior* [25; 39]. In this formulation, the optimal meta RL policy is the *Bayes-optimal policy* – the policy that maximizes expected return, where the expectation is taken with respect to the prior MDP distribution.

36th Conference on Neural Information Processing Systems (NeurIPS 2022).

While significant empirical progress in meta RL has been made, the theoretical understanding of the problem is still limited. A central question, which we focus on in this work, is the probably approximately correct (PAC) analysis of meta RL, namely, *how many training tasks are required to guarantee performance that is approximately Bayes-optimal with high probability*. Practical motivation for studying this question includes the lifelong learning setting [8], where training tasks are equivalent to tasks that the agent had previously encountered, and we seek agents that learn as quickly as possible, and the offline meta RL setting [3], where training data is collected in advance, and therefore estimating how much data is needed is important.

Recently, Tamar et al. [33] proposed the first PAC bounds for meta RL, using a *model free* approach. In their work, a history-dependent policy was trained to optimize the return on the set of training MDPs, where *policy-regularization* was added to the loss of each MDP in the data. The bounds in [33] scale with the number of states exponentiated by the length of the history, intuitively due to the number of possible histories that can be input to the policy. In this work, we propose a *model based* approach for PAC meta RL. Our idea is that instead of regularizing the policy during training, as in [33], we can learn a regularized version of the *distribution* of MDPs from the training data. Subsequently, we can train an agent to be Bayes optimal on this learned distribution. Intuitively, if we can guarantee that the learned distribution is 'close enough' to the real prior, we should expect the learned policy to be near Bayes optimal. Here, we derive such guarantees using techniques from the kernel density estimation (KDE) literature [29].

Building on PAC results for KDE, we derive PAC bounds for our model based meta RL approach. Compared to the bounds in [33], our results require much less stringent assumptions on the MDP prior, and apply to both continuous and discrete state and action spaces. Interestingly, our bounds have a linear dependence on the horizon, compared to the exponential dependence in [33], but have an exponential dependence on the dimension of the prior distribution, corresponding to the exponential dependence on dimensionality of KDE. We argue, however, that in many practical cases the prior distribution lies in some low dimensional subspace. For such cases, we extend our analysis to use dimensionality reduction via principal component analysis (PCA), and obtain bounds where the exponential dependence is on the dimension of the low dimensional subspace.

To visualize the advantage of our approach, consider the HalfCircle domain in Figure 1, adapted from [3]: a 2-dimensional agent must navigate to a goal, located somewhere on the half-circle. A task therefore corresponds to the goal location, and the task distribution is uniform on the 1-dimensional half-circle. The bounds in [33] depend on the number of states and actions, which are continuous here, and even if discretized, would result in excessive bounds. Intuitively, however, it is the low-dimensional structure of the task distribution that should determine the difficulty of the problem, and our bounds can indeed account for such. We remark that many benchmark meta RL task distributions exhibit similar low dimensional structure [4; 7; 3].

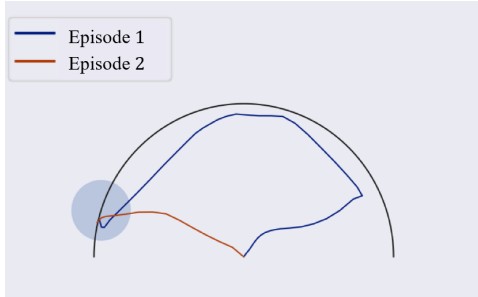

Figure 1: The HalfCircle domain (taken from [3]): the task is to navigate to a goal position that can be anywhere on the half-circle (light blue). A Bayes-optimal agent first searches along the half circle for the goal, and once found, moves directly towards it.

To complement our theoretical results, we demonstrate the practical potential of our approach by incorporating it in the VariBAD meta-RL method of Zintgraf et al. [39]. In VariBAD, a variational autoencoder (VAE) is used to learn a low-dimensional latent representation of the task distribution, and a deep neural network is trained to approximate the Bayes optimal policy. We show that by estimating the task distribution using our kernel density estimation method, applied to the VAE latent space, and training the policy on sampled tasks from this distribution, we improve the generalization of the policy to tasks not seen in the training data.

## 2 Background

We specify our notation, and background on MDPs, density estimation, and dimensionality reduction. **Notations:** Estimators will be denoted with $\widehat{\times}$. $\|\cdot\|$ denotes the Euclidean norm, other norms will be

denoted explicitly. For a set $X$, we denote by $\mathcal{P}(X)$ the set of distributions over $X$. We denote by $v_d$ the volume of a unit ball in $\mathbb{R}^d$. For a space $\mathcal{S}$ we denote $|\mathcal{S}|$ as the volume of the space.

## 2.1 Markov Decision Processes

We follow the formal setting of [33]. A Markov Decision Process (MDP) $M$ is a 7-tuple $M = (\mathcal{S}, \mathcal{A}, \mathcal{C}, P_{\text{init}}, C, P, H)$, where $\mathcal{S}$, $\mathcal{A}$ and $\mathcal{C}$ are the state, action and cost spaces, respectively, $P_{\text{init}}$ is the initial state distribution, $C : \mathcal{S} \times \mathcal{A} \to \mathcal{P}(\mathcal{C})$ is the cost function, $P : \mathcal{S} \times \mathcal{A} \to \mathcal{P}(\mathcal{S})$ is the transition function and $H \in \mathbb{Z}^+$ is the horizon. We denote by $P(c, s' \mid s, a) = P(s' \mid s, a) C(c \mid s, a)$ the probability of transitioning from state $s$ given action $a$ to state $s'$ with a cost of $c$, $\forall s, s' \in S, \forall a \in A$ and $\forall c \in \mathcal{C}$. When it is not clear from the context, we add the subscript $\times_M$ for the variables in the 7-tuple to indicate that they correspond to the particular MDP $M$. We denote the space of MDPs as $\mathcal{M}$, and use the term MDP and task interchangeably. An agent interacts with an MDP at discrete time steps: the state at time $t$ is $s_t$, where $s_0 \sim P_{\text{init}}$, the agent chooses action $a_t$, and then the state transitions to $s_{t+1} \sim P(\cdot|s_t, a_t)$ and cost $c_t \sim C(\cdot|s_t, a_t)$ is observed. We denote the history at time $t$ as $h_t = \{s_0, a_0, c_0, s_1, a_1, c_1 \ldots, s_t\}$, and the space of possible histories at time $t$ as $\mathcal{H}_t$. The agent acts based on a history-dependent policy $\pi : \mathcal{H}_t \to \mathcal{P}(\mathcal{A})$. Every $H$ steps, the episode ends and the state is reset, meaning it is sampled again from $P_{\text{init}}$. The agent's goal in an MDP is to minimize the expected cumulative cost $\mathbb{E}_{\pi, M}\left[\sum_{t=0}^{T-1} c_t\right]$. Note that the MDP horizon $H$ and the horizon $T$ of the cumulative cost might be different – this is important for our subsequent development, where the agent will face an unknown MDP, and can therefore improve its performance between episodes.

## 2.2 Kernel Density Estimation

Density estimation is the approximation of an unknown probability density function $f(x)$, $x \in \mathbb{R}^d$, based on $n$ i.i.d. samples from it $\{X_i\}_{i=1}^n$. One popular approach to density estimation is kernel density estimation (KDE [29]). Let $K(x)$ denote a kernel function, which is a probability measure over $\mathbb{R}^d$. Let $\mathbf{H_0}$ denote a $d \times d$ positive definite and symmetric matrix satisfying $det(\mathbf{H_0}) = 1$, referred to as a unit bandwidth matrix. Following the formulation of Jiang [16], we define the KDE with bandwidth $h > 0$ as follows: $\widehat{f}_{\mathbf{H_0}, h}(x) = \frac{1}{n \cdot h^d} \sum_{i=1}^{n} K\left(\frac{\mathbf{H_0}^{-1/2}(x - X_i)}{h}\right)$. Intuitively, $h$ controls the bandwidth size, while $\mathbf{H_0}$ allows for asymmetric bandwidth along different dimensions. We make this clearer with a simple example:

**Example 1.** *[Symmetric Gaussian KDE] For a Gaussian kernel function: $K(x) = \frac{1}{\left(\sqrt{2\pi}\right)^d} e^{-\frac{1}{2}x^T x}$, bandwidth matrix $\mathbf{H_0} = \mathbf{I}$, and bandwidth $h$, we obtain the Gaussian KDE: $\widehat{f}_G(x) := \frac{1}{n\left(h\sqrt{2\pi}\right)^d} \sum_{i=1}^n e^{-\frac{1}{2h^2}(x-X_i)^T(x-X_i)}$. This popular instance of the KDE can be understood as placing a Gaussian distribution with zero mean and standard deviation $h$ over each sample $X_i$.*

Jiang [16] provided finite sample bounds for the $L_\infty$ norm of the KDE error under mild assumptions.

**Assumption 1.** $\|f\|_\infty < \infty$

**Assumption 2.** *The kernel function is spherically symmetric and non-increasing, meaning there exists a non-increasing function $k : \mathbb{R}_{\geq 0} \to \mathbb{R}_{\geq 0}$ such that $K(x) = \kappa(\|x\|)$ for $x \in \mathbb{R}^d$. Further, the kernel function decays exponentially, meaning there exists $\rho, C_\rho, t_0 \in \mathbb{R} > 0$ such that $\kappa(t) \leq C_\rho \cdot \exp\left(-t^\rho\right), \forall t > t_0$.*

Notice that the Gaussian kernel in Example 1 satisfies Assumption 2.

**Assumption 3.** *$f$ is $\alpha$-Hölder continuous, meaning there exists $0 < \alpha \leq 1$, $C_\alpha > 0$, such that $|f(x) - f(x')| \leq C_\alpha \|x - x'\|^\alpha, \forall x, x' \in \mathbb{R}^d$.*

**Theorem 1.** *[Theorem 2 of Jiang [16]] Under Assumptions 1 - 3, there exists a positive constant $C' \equiv C'(d, \|f\|_\infty, C_\alpha, \alpha, K)$ such that the following holds with probability at least $1 - 1/n$, uniformly in $h > (\log n/n)^{1/d}$ and valid unit bandwidths matrices $\mathbf{H_0}$:*

$$\left\|\widehat{f}_{\mathbf{H_0}, h}(x) - f(x)\right\|_\infty < C' \cdot \left(\frac{h^\alpha}{\sigma_{min}^{\alpha/2}} + \sqrt{\frac{\log n}{n \cdot h^d}}\right), \tag{1}$$

*where $\sigma_{min}$ is the smallest eigenvalue of $\mathbf{H_0}$.*

The first term on the right hand side of (1) is a bias term, which can be reduced by reducing $h$. However, the second term will grow when $h$ is reduced. In general, the sample complexity under an optimal bandwidth scales exponentially in the dimension $d$ (see Lemma 4 for a specific example).

## 2.3 Principal Component Analysis

Principal component analysis (PCA [1; 27]) is a popular linear dimensionality reduction technique. For a $d$-dimensional random variable $X$, consider $n$ i.i.d. samples of $X$, denoted $\{X_i\}_{i=1}^n$. Reducing the dimension from $d$ to $d'$ is achieved by projection – multiplying a vector in $\mathbb{R}^d$ by a rank $d'$ orthogonal matrix $P$. The expected projection error of $P$ is $R(P) := \mathbb{E}\left[\|X - PX\|^2\right]$. Similarly, the empirical projection error of $P$ is $R_N(P) := \sum_{i=1}^n \left[\|X_i - PX_i\|^2\right]$. We denote by $\Sigma$ and $\widehat{\Sigma}$ the covariance and empirical covariance of $X$, respectively. We further denote by $\lambda_1 \geq \lambda_2 \geq \cdots \lambda_d > 0$ the eigenvalues of $\Sigma$, and by $\widehat{\lambda}_1 \geq \widehat{\lambda}_2 \geq \cdots \widehat{\lambda}_d > 0$ the eigenvalues of $\widehat{\Sigma}$.

The PCA projection $\widehat{P}_{d'}$ is constructed from the eigenvectors corresponding to the $d'$ largest eigenvalues of $\widehat{\Sigma}$. Its theoretical risk is: $\delta_{d'}^{PCA} = \mathbb{E}\left[R\left(\widehat{P}_{d'}\right)\right] - \min_{P \in \mathcal{P}_{d'}} R(P)$, where $\mathcal{P}_{d'}$ is the space of $d \times d$ orthogonal matrices with rank $d' \leq d$, and the expectation is with respect to the $n$ samples. Reiß and Wahl [27] bound the risk for sub-Gaussian random variables.

**Assumption 4.** *$X$ is sub Gaussian, meaning that its second moment is finite and there exists a constant $C_{sg}$ such that* $\sup_{k \geq 1} k^{-1/2}\mathbb{E}\left[|X \cdot u|^k\right]^{1/k} \leq C_{sg}\mathbb{E}\left[(X \cdot u)^2\right]^{1/2}, \forall u \in \mathbb{R}^d$ .[1]

It is worth mentioning that any bounded random variable satisfies Assumption 4.

**Theorem 2.** *[Proposition 2.2 of Reiß and Wahl [27]] for a random variable $X$ of dimension $d$ that satisfies Assumption 4, we have that* $\delta_{d'}^{PCA} \leq \min\left(\frac{8C_{sg}^2\sqrt{d'}tr(\Sigma)}{\sqrt{n}}, \frac{64C_{sg}^4 tr^2(\Sigma)}{n\left(\lambda_{d'} - \lambda_{d'+1}\right)}\right)$ .

# 3 Problem Statement

In this paper we make use of a parametric representation of MDPs. We suppose there is some parametric space $\Theta$ and a mapping $g : \Theta \to \mathcal{M}$, which maps parameters to MDPs. We next give two examples of such parametrizations, which will be referred to throughout the text to illustrate properties of our analysis.

**Example 2.** *[Tabular mapping] Consider the case where $\mathcal{S}, \mathcal{A}, T, \mathcal{C}$ are fixed and finite, while only $C$ and $P$ differ between different MDPs. The parametric space is defined by $\Theta = \Theta_C \times \Theta_P$, where $\Theta_C \subset \mathbb{R}^{|\mathcal{S}| \times |\mathcal{A}| \times |\mathcal{C}|}$, and $\Theta_P \subset \mathbb{R}^{|\mathcal{S}| \times |\mathcal{A}| \times |\mathcal{S}|}$, such that $\forall \theta_P \in \Theta_P, \forall \theta_C \in \Theta_C, \forall a \in \mathcal{A}$ and $\forall s \in \mathcal{S}$: $\theta_P(a, s, \cdot)$ is on the $|\mathcal{S}|$-simplex and $\theta_C(a, s, \cdot)$ is on the $|\mathcal{C}|$-simplex. The parametric mapping is defined such that: $P(s'|s, a) = \theta_P(s, a, s')$ and $C(c|s, a) = \theta_C(s, a, c)$.*

The tabular mapping in Example 2 does not assume any structure of the MDP space, and is common in the Bayesian RL literature, e.g., in the Bayes-adaptive MDP model of Duff [5]. The next example considers a structured MDP space, and is inspired by the domain in Figure 1.

**Example 3.** *[Half-circle 2D navigation] Consider the following: $\mathcal{S}, \mathcal{A} = \mathbb{R}^2$, and $P(a, s, s') = 1$ for $s' = s + a \; \forall s \in \mathcal{S}, \forall a \in \mathcal{A}$. The parameter space is $\Theta = [0, \pi]$. The parametric mapping $C(a, s) = 1, \forall a \in \mathcal{A}, \forall s \in \mathcal{S}$ such that $\|s - [R\cos\theta, R\sin\theta]\|_2 \leq r$, where $r, R \in \mathbb{R}$ .*

## 3.1 The Learning Problem

We first define the cumulative cost for a history-dependent policy $\pi$ and an MDP $M \in \mathcal{M}$ as $L_{M,\pi} = \mathbb{E}_{\pi,M}\left[\sum_{t=0}^{T-1} c_t\right]$ . We follow the Bayesian RL (BRL) formulation [9], and assume a prior distribution over the MDP parameter space $f \in \mathcal{P}(\Theta)$. Our objective is the expected cumulative cost over a randomly sampled MDP from the prior:

$$\mathcal{L}_f(\pi) = \mathbb{E}_{\theta \sim f}\left[L_{g(\theta),\pi}\right] = \mathbb{E}_{\theta \sim f}\left[\mathbb{E}_{\pi,M=g(\theta)}\left[\sum_{t=0}^{T-1} c_t\right]\right],$$

---

[1]This definition is equivalent to the standard sub-Gaussian definition, $P(|X| > t) \leq \exp(1 - t^2/\tilde{C}_{sg})$, where $C_{sg}$ and $\tilde{C}_{sg}$ differ by a universal constant [36].

where we used the parametric representation introduced above. A policy $\pi_{BO} \in \arg\min_\pi \mathcal{L}_f(\pi)$ is termed *Bayes optimal*. If the prior distribution is known, one can calculate the Bayes optimal policy [9]. However, in our setting we assume that $f$ is not known in advance, motivating the following learning problem.

We are given a training set of MDPs, $\{\theta\}_{i=1}^N$, sampled independently from the prior $f(\theta)$. Our goal is to use these MDPs to calculate a policy that minimizes the regret:

$$\mathcal{R}(\pi) = \mathcal{L}_f(\pi) - \mathcal{L}_f(\pi_{\mathrm{BO}}) = \mathbb{E}_{\theta \sim f}\left[\mathbb{E}_{\pi, M=g(\theta)}\left[\sum_{t=0}^{T-1} c_t\right] - \mathbb{E}_{\pi_{\mathrm{BO}}, M=g(\theta)}\left[\sum_{t=0}^{T-1} c_t\right]\right]$$

Note that the expectation above is with respect to the true prior, therefore, the regret can be interpreted as how well the policy calculated from the training data *generalizes* to an unseen test MDP.

## 4   Generalization Bounds

We next provide PAC bounds for the learning problem of Section 3. Our general idea is to first estimate the prior distribution $f(\theta)$ from the training set using KDE, and then solve for the Bayes optimal policy with respect to the KDE-estimated distribution instead of the real prior. For an estimator $\widehat{f}(\theta)$ of the real prior $f(\theta)$, we define the estimated Bayes optimal policy: $\pi^*_{\widehat{f}} \in \arg\min_\pi \mathcal{L}_{\widehat{f}}(\pi)$.

We start by showing that we can bound the regret of an estimated Bayes optimal policy, as a function of the estimation error of the prior itself. The proof, detailed in Section A.2 in the supplementary, is a simple application of norm inequalities, and exploiting the fact that the total cost is bounded.

**Lemma 3.** *Let $\widehat{f} \in \mathcal{P}(\mathbb{R}^d)$ be an estimator of the real prior $f$ over the parametric space $\Theta$. We have that: $\mathcal{R}(\pi^*_{\widehat{f}}) = \mathcal{L}_f(\pi^*_{\widehat{f}}) - \mathcal{L}_f(\pi_{\mathrm{BO}}) \leq 2C_{max}T\left\|f - \widehat{f}\right\|_1$, and for a bounded parametric space of volume $|\Theta|$ we have: $\mathcal{R}(\pi^*_{\widehat{f}}) = \mathcal{L}_f(\pi^*_{\widehat{f}}) - \mathcal{L}_f(\pi_{\mathrm{BO}}) \leq 2C_{max}T|\Theta|\left\|f - \widehat{f}\right\|_\infty$.*

While the assumption of a finite parametric space volume is reasonable in practice, the volume size, appearing in the $\|\cdot\|_\infty$ bound that we shall use in the proceeding analysis, grows exponentially with the dimension of $\Theta$. In Section 4.1 we will discuss how, under some assumptions, we can relax this.

Lemma 3 provides a convenient framework for bounding the regret using any density estimation technique with a known bound. We now consider the special case of the KDE (cf. Section 2.2) with a Gaussian kernel function and an optimal selection of bandwidth, as calculated in the following lemma. We focus on the Gaussian kernel both for simplicity and because it is a popular choice in practice. Our method can be extended to any kernel function that satisfies assumption 2.

**Lemma 4.** *The optimal KDE bandwidth is (up to a constant independent of $n$) $h^* = \left(\log n / n\right)^{\frac{1}{2\alpha+d}}$.*

The following result bounds the estimation error for this case. The proof, detailed in Section A.4 of the supplementary material, is based on Theorem 1 from [16], where the constants are calculated explicitly for the Gaussian kernel case.

**Lemma 5.** *Under Assumptions 1 and 3, for a parametric space with finite volume $|\Theta|$ and a KDE with a Gaussian kernel $K(u) = \dfrac{e^{-\frac{1}{2}u^T u}}{(2\pi)^{\frac{d}{2}}}$, $\mathbf{H_0} = \mathbf{I}$, and an optimal bandwidth $h^*$, we have that with probability at least $1 - 1/n$: $\sup_{x\in\mathbb{R}^d}\left|\widehat{f}_G(x) - f(x)\right| \leq C_d \cdot \left(\frac{\log n}{n}\right)^{\frac{\alpha}{2\alpha+d}}$, where $C_d = C_\alpha 2^{\frac{\alpha-1}{2}} + \dfrac{16d\sqrt{C_\alpha \Delta_{max}^\alpha(\Theta) + \frac{1}{|\Theta|}}}{\sqrt{2}(2\pi)^{\frac{d}{4}}} + \dfrac{64d^2}{(2\pi)^{\frac{d}{2}}}$, and $\Delta_{max}(\Theta)$ is the maximal $L_1$ distance between any two parameters in $\Theta$.*

**Remark 1.** *Usually, $\Delta_{max}(\Theta)$ will scale polynomially (or sub-polynomially) with $d$. For example, in case the parametric space is a $d$-dimensional hypercube with edge length $B$, we have: $\Delta_{max} = \sqrt{d}B$. One parametric case that can be represented as a hyper cube is Example 2, where the edge length is $B = 1$, and $d = |\mathcal{S}|^2|\mathcal{A}| + |\mathcal{S}||\mathcal{A}||\mathcal{C}|$. In such cases, for large enough $d$, the first term in the equation for $C_d$ will dominate, and therefore $C_d \approx C_\alpha 2^{\frac{\alpha-1}{2}}$.*

By combining Lemma 3 and 5 we obtain a regret bound – the main result of this section.

**Theorem 6.** *For a prior $f(\theta)$ over a bounded parametric space $\Theta$ that satisfies Assumptions 1 and 3, and a Gaussian KDE with optimal bandwidth, we have that with probability at least $1 - 1/n$:*

$$\mathcal{R}_T(\pi^*_{\widehat{f}_G}) \leq 2C_{max}T \, |\Theta| \, C_d \cdot \left(\frac{\log n}{n}\right)^{\frac{\alpha}{2\alpha+d}} .$$

**Remark 2.** *While Theorem 6 assumes a bounded parametric space, the resulting KDE estimate is not necessarily bounded (e.g., when using a Gaussian kernel). In Section A.5 of the supplementary, we show that the result of Theorem 6 also holds when truncating the KDE estimate to a support $\Theta$.*

We next compare Theorem 6 with the the bounds of Tamar et al. [33]. We recall that [33] considered a model-free approach that learns a history-dependent policy $\widehat{\pi}^*_{reg}$ on the the training domains, with $L_2$ policy regularization. Tamar et al. [33] only considered finite state, action, and cost spaces, a setting equivalent to our tabular representation in Example 2, where the parametric space dimension is $d = |\mathcal{S}|^2 |\mathcal{A}| + |\mathcal{S}| |\mathcal{A}| |\mathcal{C}|$. Corollary 1 in [33] shows that with probability at least $1 - 1/n$,
$\mathcal{R}_T\left(\widehat{\pi}^*\right) \leq 2\sqrt{\rho} \cdot n^{-\frac{1}{4}}T + 2\sqrt{\rho}n^{-\frac{3}{4}} + \left(\frac{4\sqrt{\rho}}{n^{-\frac{1}{4}}} + 3C_{\max}T\right)\left(\frac{\log n}{2n}\right)^{\frac{1}{2}}$, where $\rho = 2q^{2T}C_{\max}^2 T^2 |\mathcal{A}|$, and $q = \sup_{M,M'\in\mathcal{M}, s,s'\in\mathcal{S}, a\in\mathcal{A}, c\in\mathcal{C}} P_M(s',c|s,a)/P_{M'}(s',c|s,a)$.[2]
At first sight, the exponent $\frac{\alpha}{2\alpha+d}$ in our bound compared to the $\frac{1}{4}$ in [33] is significantly worse. The intuitive reason for this is that Theorem 6 builds on KDE, where it is natural to expect an exponential dependence on the dimension $d$ when estimating the density $f(\theta)$. Yet, one must also consider the constants. Assuming that $q$ is finite places a severe constraint on the space of possible MDPs – essentially, each cost and transition must be possible in all MDPs in the prior! Tamar et al. [33] claim that $q$ may be made finite by adding small noise to every transition, but in this case $q$ becomes $O(|\mathcal{S}||\mathcal{C}|)$, leading to an exponential $O((|\mathcal{S}||\mathcal{C}|)^T)$ term in the bound of [33], while our bound scales linearly with $T$. The intuitive reason for this exponential dependence on $T$ is that when learning a history-dependent policy, the space of possible histories grows exponentially with $T$.

To summarize, for a small $T$, and when the structure of $\mathcal{M}$ is such that $q$ is finite, the model-free approach of [33] seems preferable. However, when the dimension $d$ is small, and $T$ is large, our model based approach has the upper hand.

**Remark 3.** *Tamar et al. [33] also considered a case where $\mathcal{M}$ is a finite set of size $|\mathcal{M}|$, and in this case, their Corollary 2 shows that for $0 < \alpha < 1$, with probability $1 - 1/n^\alpha$, $\mathcal{R}_T\left(\widehat{\pi}^*\right) \leq \left(2 + \sqrt{\frac{48C_{max}^3|\mathcal{A}|}{2P_{\min}}}\right)T^{\frac{4}{3}}n^{-\frac{1-\alpha}{3}}$, where $P_{min} = \min_{M\in\mathcal{M}} P(M)$.[3] In the case of a discrete and finite parametric space there is no need for the KDE, but we can still use our model based approach, and estimate the prior using the empirical distribution $\widehat{P}_{emp}(M) = \widehat{n}(M)/n, \forall M \in \mathcal{M}$, where $\widehat{n}(M)$ is the number of occurrences of $M$ in the training set. We can bound the $L_1$ error of this estimator using the Bretagnolle-Huber-Carol inequality [35] (the full proof is outlined in Section A.6 in the supplementary), and obtain that with probability at least $1 - 1/n^\alpha$, we have that $\mathcal{R}_T(\pi^*_{\widehat{P}_{emp}}) \leq 2C_{max}T\sqrt{2\left(\alpha \log\left(n \log 2\right) + |\mathcal{M}| + 1\right)/n}$. Observe that the dependence on $T$ and $C_{max}$ in this bound is better than in the bound of [33], and we do not have the $P_{min}$ term in the denominator, which could be very small if some MDPs in the prior are rare, and is at most $1/|\mathcal{M}|$. Ignoring the $\log$ term, our dependence on $n$ is also better for every $\alpha > 0$.*

We conclude this section by pointing out that, as exemplified in Remark 3, our approach can be generalized to density estimation techniques beyond KDE, so long as their error can be bounded.

### 4.1 Bounds for Parametric Spaces with Low Dimensional Structure

The sample complexity in the general bound of Theorem 6 grows exponentially with the dimension of the parameter space $\Theta$. In many practical cases however, such as the HalfCircle domain of Example 3, there may be a low dimensional representation that encodes most of the important information in the tasks, even though the dimensionality of the parametric space is higher. In such cases, we expect that our bounds can be improved to depend on the *low dimensional* representation. In the following, we

---

[2]The bound in [33] also has a term $\lambda$ for the weight of the $L_2$ regularization term. Here, we present the result for the optimal $\lambda$, which can be calculated similarly to the optimal bandwidth in Lemma 4.

[3]We present the result of [33] for the optimal regularization coefficient, and ignore a subleading $n^{-\frac{1-a}{2}}$ term.

approach this task by combining our model-based approach with the PCA dimensionality reduction method. PCA is a linear method, and allows for a relatively simple analysis to demonstrate our claim. In practice, a non-linear method may be preferred. In Section 6, we verify empirically that our approach also works with non-linear deep neural network based dimensionality reduction.

We propose the following procedure.

1. Reduce the training set dimensionality from $d$ to $d'$ using PCA
2. Perform a Gaussian KDE estimation with optimal bandwidth in the low dimension $d'$
3. Project back the estimated distribution to dimension $d$, to obtain the prior estimator $\widehat{f}_G^{d'}$

The main difficulty in analysing this procedure, however, is calculating the error of the estimated distribution in dimension $d$, that is, *after the projection step*. We remark that projecting back is necessary, as calculating the estimated Bayes optimal policy $\pi_{\widehat{f}}^*$ requires $g(\theta)$, where $\theta$ is in dimension $d$. To set the stage, we first need to define the projection step explicitly.

The PCA dimensionality reduction can be written as $\widehat{P}_{d'} = W_L^T \cdot W_L$, where $W_L \in \mathbb{R}^{d' \times d}$. For any $\theta \in \Theta$, we therefore have that $\theta_L = W_L \cdot \theta$ is the low dimensional representation of $\theta$, and $W_L^T \cdot \theta_L$ is the projection of $\theta_L$ back to the $d$-dimensional space. For each $\theta_L$, we denote its inverse image as follows: $\Theta_L^\perp(\theta_L) = \{\theta \in \Theta : W_L \cdot \theta = \theta_L\}$. By the law of total probability, the probability distribution of a low-dimensional $\theta_L$ is: $f_{\theta_L}(\theta_L) = \int_{\theta_L^\perp \in \Theta_L^\perp(\theta_L)} f_\theta(\theta_L^\perp) d\theta_L^\perp$.

For the proceeding analysis, we require the function that maps parameters to MDPs to be smooth.

**Assumption 5.** *In the MDP space $\mathcal{M}$, only $P$ and $C$ can differ. Furthermore, the parametric mapping is Lipshitz continuous with respect to $P_M(\cdot, \cdot \mid s, a)$, i.e: $\exists C_g$ such that $\forall s \in \mathcal{S}, a \in \mathcal{A}$, $\left\| P_{M=g(\theta_1)}(\cdot, \cdot \mid s, a) - P_{M=g(\theta_2)}(\cdot, \cdot \mid s, a) \right\|_1 \le C_g \left\| \theta_1 - \theta_2 \right\|_1.$*

We now extend the well known *Simulation Lemma* [17] to the case of a history-dependent policy. This will allow us later to relate the error in the prior distribution to the error of the policy.

**Lemma 7.** *For any history-dependent policy $\pi$ and any parametric mapping $g$ that satisfies Assumption 5, the following holds for any $\theta_1, \theta_2 \in \Theta$:*

$$\left| L_{M=g(\theta_1), \pi} - L_{M=g(\theta_2), \pi} \right| \le C_{max} C_g \left\| \theta_1 - \theta_2 \right\|_1 \cdot T^2. \tag{2}$$

We note that Assumption 5 measures closeness between the *joint* distribution over costs and transitions. This is slightly different than the simulation lemma for a Markov policy [17], where only the absolute difference between the costs is required. Intuitively, this is since a history-dependent policy can depend on observed costs, therefore even if two observed costs are very close in magnitude, the actions resulting from observing them could be very different. We note that in the case where $P$ and $C$ are bounded (which is always true when $\mathcal{S}, \mathcal{A}$ and $\mathcal{C}$ are discrete), it is enough to assume Lipschitz continuity of $g$ with respect to $P$ and $C$ separately. We also note that there may be parameter spaces that satisfy Eq. 2 without satisfying Assumption 5; in such cases our proceeding results will still hold.

In the following, we would like to formally consider MDP spaces with a low-dimensional structure. This is captured by assuming that the MDPs essentially lie on a linear subspace of dimension $d'$, such that the magnitude of the variability of MDPs outside this subspace is bounded by $\epsilon$.

**Assumption 6.** *There exist $d' \le d$ and $\epsilon \in \mathbb{R}_{\ge 0}$ such that $\lambda_{d'+1} \le \epsilon$, where $\lambda_i$, as defined in Section 2.3, is the $i$'th largest eigenvalue of the Covariance matrix of $\theta$.*

Under Assumption 6, and using the PCA bounds of Reiß and Wahl [27] (cf. Theorem 2), we can bound the dimensionality reduction error due to performing PCA on *the sampled* MDPs in our data, as given by the following lemma.

**Lemma 8.** *Under Assumptions 4 and 6, we have that:*

$$\mathbb{E}\left[ R\left( \widehat{P}_{d'} \right) \right] \le \min\left( \frac{8 C_{sg}^2 \sqrt{d'} tr(\Sigma)}{\sqrt{n}}, \frac{64 C_{sg}^4 tr^2(\Sigma)}{n \left( \lambda_{d'} - \lambda_{d'+1} \right)} \right) + \epsilon \cdot (d - d').$$

There are two terms to the bound in Lemma 8. The first is due to the error in performing PCA with a finite number of samples, and decays as $n$ increases. The second is due to the fact that even for a perfect PCA there is some error, as the MDPs do not lie perfectly in a low dimensional subspace.

We next assume that after the PCA projection, the distribution remains Hölder continuous.

**Assumption 7.** $f_{\theta_L}$ *is* $\alpha'$-*Hölder continuous, meaning there exists* $0 < \alpha' \leq 1$, $C_{\alpha'} > 0$, *such that* $|f(\theta_L) - f(\theta'_L)| \leq C_{\alpha'} \|\theta_L - \theta'_L\|^{\alpha'}$, $\forall \theta_L, \theta'_L \in \Theta_L$.

**Theorem 9.** *Let* $\widehat{f}_G^{d'}$ *be the approximation of* $f$ *as defined above. Under Assumptions 1, 4, 6, and 7 we have with probability at least* $1 - 1/n$:

$$\mathcal{R}_T\left(\pi^*_{\widehat{f}_G^{d'}}\right) \leq 2C_{max}T|\Theta_L|C_{d'}\left(\frac{\log n}{n}\right)^{\frac{\alpha'}{2\alpha'+d'}} + 2C_{max}T^2C_g\sqrt{\min\left(\frac{C'_{sg}\sqrt{d'}}{\sqrt{n}}, \frac{C'^2_{sg}}{n\Delta_{\lambda,d'}}\right)} + \epsilon(d - d'),$$

*where* $C_{d'} = C_{\alpha'}2^{\frac{\alpha'-1}{2}} + \frac{16d'\sqrt{C_{\alpha'}\Delta^{\alpha'}_{max}(\Theta_L)+\frac{1}{|\Theta_L|}}}{\sqrt{2}(2\pi)^{\frac{d'}{4}}} + \frac{64d'^2}{(2\pi)^{\frac{d'}{2}}}$, $C'_{sg} = 8C^2_{sg}tr(\Sigma)$, *and* $\Delta_{\lambda,d'} = \lambda_{d'} - \lambda_{d'+1}$.

The first term in the bound of Theorem 9 is the KDE error. Note that, compared to the KDE error in Theorem 6, the exponential dependence is on *the low dimension* $d'$, and not on the higher dimension $d$. The second term in the bound is due to the PCA error, as discussed after Lemma 8. This result demonstrates the potential in our approach, which accounts for structure in the parametric space.

## 5   Related Work

Meta RL has seen extensive empirical study [4; 39; 7; 26], and the connection to Bayesian RL has been made in a series of recent works [20; 14; 25; 39]. Most meta RL studies assumed infinite tasks during training (effectively drawing different random MDPs from the prior at each training iteration), with few exceptions, mostly in the offline meta RL setting [3; 22].

Bayesian RL algorithms such as [12; 10] sample MDPs from the true posterior, which requires knowing the true prior. In their comprehensive study, Simchowitz et al. [32] analyse a family of posterior-sampling based algorithms, termed $n$-Monte Carlo methods, under mis-specifed priors, and bound the corresponding difference in accumulated regret. In contrast, our work considers a zero-shot setting, comparing the Bayes-optimal policies with respect to the true and the mis-specifed prior, and, more importantly, focuses on the sample complexity of *obtaining* a prior that is accurate enough – an issue that is not addressed in [32] for MDPs. To the best of our knowledge, the only theoretical investigation of meta RL with finite training tasks in a similar setting to ours is the model free approach in [33], which we compare against in our work. Our theoretical analysis builds on ideas from the study of density estimation [29; 16] and dimensionality reduction [1; 27].

We note that regularization techniques inspired by the mixup method [38] have been applied to meta learning [37], and recently also to meta RL [21], with a goal of improving generalization to out-of-distribution tasks. In our experiments, we compared our approach with an approach inspired by [21], and found that for in-distribution generalization, our approach worked better. That said, we believe there is much more to explore in developing effective regularization methods for meta RL. We conclude with studies on PAC-Bayes theory for meta learning [2; 30; 6]. These works, which have a different flavor from our PAC analysis, do not cover the meta RL problem considered here.

## 6   Experiments

In this section we complement our theoretical results with an empirical investigation. Our goal is to show that our main idea of learning a KDE over a low dimensional space of tasks is effective also for state-of-the-art meta-RL algorithms, for which the linearity assumption of PCA clearly does not hold, and computing the optimal yet intractable $\pi^*_{\widehat{f}}$ is replaced with an approximate deep RL method.

Modern deep RL algorithms are known to be highly sensitive to many hyperparameters [13], and meta RL algorithms are not different. To demonstrate our case clearly, we chose to build on the VariBAD algorithm of Zintgraf et al. [39], for which we could implement our approach by replacing just a single algorithmic component, as we describe next, and thus obtain a fair comparison.

**VariBAD:** We briefly explain the VariBAD algorithm, to set the stage for our modified algorithm to follow; we refer the reader to [39] for the full details. VariBAD is composed of two main components. The first is a variational autoencoder (VAE [19]) with a recurrent neural network (RNN) encoder that, at each time step, encodes the history into a Gaussian distribution over low dimensional latent vectors. The decoder part of the VAE is trained to predict the next state and reward from the encoded history.

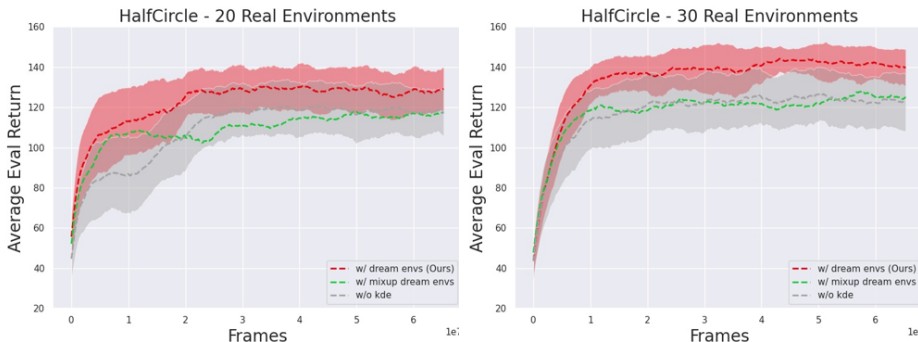

Figure 2: Average return on HalfCircle with KDE and mixup dream environments and without dream environments. The average is shown in dashed lines, with the 95% confidence intervals (15 random seeds). We do not show the intervals for the mixup run for visualization clarity; mixup obtained similar intervals as without dream. The full comparison is in Section A.11 of the supplementary.

Using the terminology in our paper, the VAE latent state can be related to the MDP parameter $\theta$, and the VAE decoder learns a model of $g(\theta)$. The second component in VariBAD is a policy, mapping the output of the VAE encoder and current state into an action. This component, which can be seen as producing an approximation of $\pi_{\hat{f}}^*$, is trained using a policy gradient algorithm such as PPO [31].

**VariBAD Dream:** Recall that our pipeline is to learn a KDE over the task parameters $\theta$, and then train a policy on tasks from the estimated KDE. Unfortunately, in our meta-RL setting, we do not assume that we directly know the $\theta$ representation for each task. However, the VAE in VariBAD allows for a convenient approximation. We can think of the output of the VAE encoder at the end of an episode, after a full history has been observed and the uncertainty about the task has been resolved as best as possible, as a sample of the task parameter $\theta$. Thus, we propose to build our KDE estimate over these variables. We henceforth refer to samples from the KDE as **dream environments** (as they are not present in the real data), and we note that the VAE decoder can be used to sample rewards and state transitions from these environments. Thus, we can train the VariBAD policy on both the sampled training environments, and also on dream environments. We refer to this method as *VariBAD Dream*. In our implementation, we train the KDE and VariBAD components simultaneously. The full implementation details and pseudo code are in Section A.10 of the supplementary.

We emphasize that in the original VariBAD work [39], the algorithm was trained by sampling different tasks from the task distribution *at each iteration*, corresponding to an infinite number of traning tasks. As we are interested in the finite task setting, our implementation of VariBAD (and VariBAD Dream) draws a single batch of $N_{train}$ tasks once, at the beginning of training, and subsequently samples tasks from within this batch for training the VAE and policy.

**Results:** We consider the HalfCircle environment of Figure 1 – a popular task that requires learning a non-trivial Bayes-optimal policy [3]. In order to test generalization, we evaluated the agents on $N_{eval} = 50$ sampled environments, different from the $N_{train}$ training ones. In Figure 2 we show the affect of incorporating the dream environments on the average return for the evaluation environments. Evidently, our approach achieved higher return, both for $N_{train} = 20$ and $N_{train} = 30$. For $N_{train} = 40$, the original VariBad performed near optimally, and there was no advantage for VariBad Dream to gain. For $N_{train} = 10$, the sampling was too sparse, and both methods demonstrated comparable failure in generalizing to the test environments. We emphasize that while the performance advantage of VariBAD Dream is modest, it is remarkable that the KDE regularization demonstrates consistent improvement, as the VAE training and PPO optimization already include significant implicit and explicit regularization mechanisms.

Additionally, we evaluated a method inspired by mixup [38; 37; 21], where dream environments are created by a random weighted average of latent vectors (instead of the KDE). Interestingly, our KDE approach outperformed this method, which may be the result of the geometry of the task distribution – there are no goals within the half circle, in contrast with the experiments in [21], where the goals were distributed *inside* a rectangular area. We remark that the experiments in [21] were designed to evaluate *out-of-distribution* generalization, different from the *in distribution* setting considered here. In Sections A.11 and A.12 of the supplementary we provide additional evaluations and show similar results on a MuJoCo [34] environment with a high dimensional state space.

**Synthetic Experiments with a Known Parametric Space:** In our theoretical analysis, we assumed knowledge of both the parameters $\theta$ of the training MDPs, and the function $g(\theta)$ that maps parameters to MDPs. We complement our empirical investigation with synthetic experiments with VariBAD Dream, where $g(\theta)$ is known.

We consider again the HalfCircle environment of Figure 1. We define the parametric space as $\Theta = \mathbb{R}^2$, where $g(\theta)$ prescribes a corresponding MDP with a goal at location $(x, y) = \theta$. For our experiment, we sample $\{\theta_i\}_{i=1}^N$ i.i.d. from the training environment distribution, and perform KDE directly on these samples. Our variant of VariBad Dream samples parameters from the KDE and generates dream MDPs using the known $g(\theta)$.

Figure 3 compares VariBAD Dream with the conventional VariBad (trained only on $\{\theta_i\}_{i=1}^N$), on unseen test environments. Evidently, the use of KDE improved the generalization performances of VariBad by a large margin, especially when the number of training environments is small. Note that the improvement is starker in this synthetic setting compared to Figure 2, as VariBAD Dream here does not suffer from inaccuracies due to learning a model of $g(\theta)$.

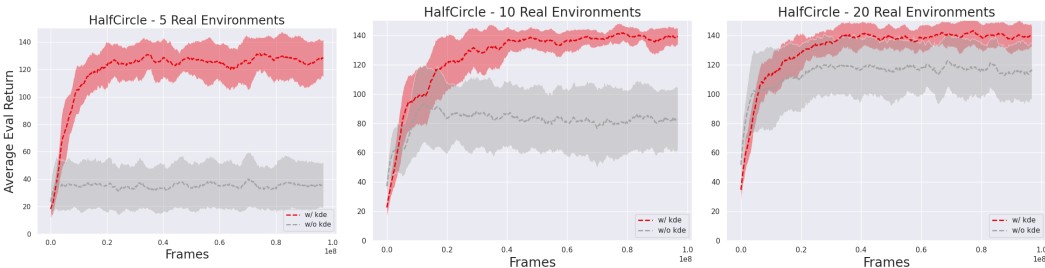

Figure 3: Average return on half-circle of the original VariBad and the VariBAD Dream variant with access to the MDP parametric space. The average is shown in dashed lines, with the 95% confidence intervals (using 6 random seeds).

## 7  Discussion & Future Work

We propose a model-based scheme for meta-RL with a finite sample of training tasks, where we first estimate the prior distribution of tasks, and train a Bayes optimal policy on the estimated prior. Using KDE for density estimation, we obtain state-of-the-art PAC bounds. Further, our approach can exploit low dimensional structure of the task distribution, when such exists, to obtain improved bounds. Finally, we showed that our approach can be "plugged-in" the VariBAD algorithm to improve generalization.

A key takeaway from our analysis is that the *dimensionality* of the task distribution determines, with exponential dependency, the amount of training samples required to act well. This insight provides a rule-of-thumb of when meta RL approaches based on task inference, such as VariBAD, are expected to work well. Indeed, recent empirical work by Mandi et al. [23] claimed that in benchmarks such as RLBench [15], where tasks are very diverse, simpler meta RL methods based on fine-tuning a policy trained on diverse tasks display state-of-the-art performance.

There are several exciting future directions for investigation. Our theoretical analysis can be extended to more advanced density estimation and dimensionality reduction techniques, such as VAEs [28]. Our empirical investigation hinted that regularization, such as affected by VariBAD Dream, can improve deep meta-RL algorithms. More sophisticated regularization can be developed based on prior knowledge about the possible tasks.

## Acknowledgements

This work received funding from Ford Inc., and from the European Union (ERC, Bayes-RL, Project Number 101041250). Views and opinions expressed are however those of the author(s) only and do not necessarily reflect those of the European Union or the European Research Council Executive Agency. Neither the European Union nor the granting authority can be held responsible for them.

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
