# A   Appendix

## A.1   Limitations

In this section we summarize several limitations of our study.

Our analysis applies to linear dimensionality reduction (PCA), while in practice, it may be that the prior lives on a low dimensional non-linear manifold. Extending our theory for such cases would require a significantly more elaborate approach. However, our experiments show that empirically, our insights still hold when the PCA is replaced with a non-linear VAE.

As discussed in Section 4, the bounds achieved by our method are exponential with the underlying dimension of the task distribution. The lower-bound example in Proposition 2 of [33] can be adapted to our setting (by using the Tabular Mapping in Example 2), to show a problem setting where an exponential dependence on the dimension cannot be avoided, regardless of the algorithm. Thus, without additional structure in the problem, this limitation is general to meta-RL and not specific to our method.

The proposed algorithm, VariBad Dream, builds on top of VariBad, and requires the latent space to be learned by the VariBad algorithm. We are therefore limited to environments in which VariBad performs adequately. Furthermore, in order to create the prior estimate using KDE, we used VariBAD latents gathered from the VAE posterior at the end of a VariBad rollout. This may not work well in some cases, e.g., when the task uncertainty is not resolved at the end of the episode.

## A.2   Regret Bounds Using Prior Estimation

**Lemma 3**. Let $\widehat{f} \in \mathcal{P}(\mathbb{R}^d)$ be an estimator of the real prior $f$ over the parametric space $\Theta$. We have that: $\mathcal{R}(\pi_{\widehat{f}}^*) = \mathcal{L}_f(\pi_{\widehat{f}}^*) - \mathcal{L}_f(\pi_{\mathrm{BO}}) \leq 2C_{max}T \left\| f - \widehat{f} \right\|_1$, and for a bounded parametric space of volume $|\Theta|$ we have: $\mathcal{R}(\pi_{\widehat{f}}^*) = \mathcal{L}_f(\pi_{\widehat{f}}^*) - \mathcal{L}_f(\pi_{\mathrm{BO}}) \leq 2C_{max}T \left| \Theta \right| \left\| f - \widehat{f} \right\|_\infty$.

*Proof of Lemma 3.*

$$\mathcal{L}_f(\pi) - \mathcal{L}_{\widehat{f}}(\pi) = \mathbb{E}_{\theta \sim f(\theta)}\mathbb{E}_{\pi, M=g(\theta)}\left[\sum_{t=0}^{T-1} c_t\right] - \mathbb{E}_{\theta \sim \widehat{f}(\theta)}\mathbb{E}_{\pi, M=g(\theta)}\left[\sum_{t=0}^{T-1} c_t\right]$$

$$= \int \mathbb{E}_{\pi, M=g(\theta)}\left[\sum_{t=0}^{T-1} c_t\right] f(\theta)d\theta - \int \mathbb{E}_{\pi, M=g(\theta)}\left[\sum_{t=0}^{T-1} c_t\right] \widehat{f}(\theta)d\theta$$

$$= \int \mathbb{E}_{\pi, M=g(\theta)}\left[\sum_{t=0}^{T-1} c_t\right] \left(f(\theta) - \widehat{f}(\theta)\right) d\theta$$

Taking the absolute value:

$$\left| \mathcal{L}_f(\pi) - \mathcal{L}_{\widehat{f}}(\pi) \right| = \left| \int \mathbb{E}_{\pi, M=g(\theta)}\left[\sum_{t=0}^{T-1} c_t\right] \left(f(\theta) - \widehat{f}(\theta)\right) d\theta \right|$$

$$\leq \int \left| \mathbb{E}_{\pi, M=g(\theta)}\left[\sum_{t=0}^{T-1} c_t\right] \right| \left| \left(f(\theta) - \widehat{f}(\theta)\right) \right| d\theta$$

$$\leq C_{max}T \int \left| \left(f(\theta) - \widehat{f}(\theta)\right) \right| d\theta \leq C_{max}T \left\| f_1 - f_2 \right\|_1$$

Using the above with $A = C_{max}T \left\| f - \widehat{f} \right\|_1$:

$$\mathcal{L}_f(\pi_{\widehat{f}}^*) - A \leq \mathcal{L}_{\widehat{f}}(\pi_{\widehat{f}}^*) \leq \mathcal{L}_{\widehat{f}}(\pi_f^*) \leq \mathcal{L}_f(\pi_f^*) + A$$

$$\Rightarrow \mathcal{L}_f(\pi_{\widehat{f}}^*) - A \leq \mathcal{L}_f(\pi_f^*) + A$$

Rearranging gives the first bound. For a finite parametric space of size $|\Theta|$, we know that $\left\| f - \widehat{f} \right\|_1 \leq |\Theta| \cdot \left\| f - \widehat{f} \right\|_\infty$, which yields the second bound. $\qquad\square$

## A.3 Optimal KDE Bandwidth

**Lemma 4.** The optimal KDE bandwidth is (up to a constant independent of $n$) $h^* = \left(\log n / n\right)^{\frac{1}{2\alpha+d}}$

*Proof of Lemma 4.* The minimum value of the function $f(x) = Ax^a + Bx^b$ with $A, a, B, b \neq 0$ is achieved with:

$$x^* = \left( -\frac{bB}{aA} \right)^{\frac{1}{a-b}}$$

We can use this result to achieve the optimal bandwidth for the bound in lemma 1, with: $A = C'\sigma_{min}^{-\alpha/2}$, $a = \alpha$, $B = C'\sqrt{\frac{\log n}{n}}$ and $b = -\frac{d}{2}$, resulting with:

$$\arg\min_{h \in \mathbb{R}^+} \left\| \widehat{f}_{\mathbf{H}}(x) - f(x) \right\|_\infty = \left( \frac{d^2 \log n}{4\alpha^2 n} \right)^{\frac{1}{2\alpha+d}}$$

$\qquad\square$

## A.4 Gaussian KDE Bounds

**Lemma 5.** Under Assumptions 1 and 3, for a parametric space with finite volume $|\Theta|$ and a KDE with a Gaussian kernel $K(u) = \frac{e^{-\frac{1}{2}u^T u}}{(2\pi)^{\frac{d}{2}}}$, $\mathbf{H_0} = \mathbf{I}$, and an optimal bandwidth $h^*$, we have that with probability at least $1 - 1/n$: $\sup_{x \in \mathbb{R}^d} \left| \widehat{f}_G(x) - f(x) \right| \leq C_d \cdot \left( \frac{\log n}{n} \right)^{\frac{\alpha}{2\alpha+d}}$, where $C_d = C_\alpha 2^{\frac{\alpha-1}{2}} + \frac{16d\sqrt{C_\alpha \Delta_{max}^\alpha(\Theta) + \frac{1}{|\Theta|}}}{\sqrt{2}(2\pi)^{\frac{d}{4}}} + \frac{64d^2}{(2\pi)^{\frac{d}{2}}}$, and $\Delta_{max}(\Theta)$ is the maximal $L_1$ distance between any two parameters in $\Theta$.

*Proof of Lemma 5.* We follow the proof of Theorem 2 in [16].

We define:

$$\check{u}_x(r) := f(x) - \inf_{x' \in B(x,r)} f(x')$$

and

$$\widehat{u}_x(r) := \sup_{x' \in B(x,r)} f(x') - f(x)$$

the following holds [16]:

$$\int_{\mathbb{R}^d} K(u)\check{u}_x \left( \frac{h\|u\|}{\sqrt{\sigma_{min}}} \right) du \leq \frac{v_d \cdot C_\alpha h^\alpha}{\sigma_{min}^{\alpha/2}} \int_0^\infty k(t) t^{d+\alpha} dt$$

and the above equation is also valid when replacing $\check{u}_x$ with $\widehat{u}_x$

We can use Theorem 1 from [16] and get:

$$\sup_{x \in \mathbb{R}^d} \left| \widehat{f}_{\mathbf{H}}(x) - f(x) \right| < \epsilon h^\alpha + C_\infty \sqrt{\frac{\log n}{n \cdot h^d}}$$

Where $\epsilon = \frac{v_d \cdot C_\alpha}{\sigma_{min}^{\alpha/2}} \int_0^\infty k(t) t^{d+\alpha} dt$, and $C_\infty = 8d\sqrt{v_d \cdot \|f\|_\infty} \left( \int_0^\infty k(t) \cdot t^{d/2} dt + 1 \right) + 64d^2 \cdot k(0)$.

$\kappa$ is the function introduced in assumption 2 and $v_d = \frac{\pi^{d/2}}{\Gamma(1+d/2)}$ is the volume of the d-dimensional unit ball, where $\Gamma$ is the Gamma function.

For KDE with the optimal bandwidth $h^* = \left(\frac{\log n}{n}\right)^{\frac{1}{2\alpha+d}}$, defined in lemma 4 we get:

$$\sup_{x\in\mathbb{R}^d} \left|\widehat{f_{\mathbf{H}}}(x) - f(x)\right| < \epsilon h^\alpha + C_\infty \sqrt{\frac{\log n}{n \cdot h^d}} = (\epsilon + C_\infty) \cdot \left(\frac{\log n}{n}\right)^{\frac{\alpha}{2\alpha+d}}$$

In the case of the Gaussian kernel presented in example 1: $K(x) = \frac{e^{-\frac{1}{2}x^T x}}{(2\pi)^{\frac{d}{2}}}$, $k(t) = \frac{e^{-\frac{1}{2}t^2}}{\sqrt{(2\pi)^d}}$ and $\sigma_{min} = 1$.

The well known formula for the moments of the Gaussian distribution:

$$\int_0^\infty k(t) \cdot t^a dt = \frac{\Gamma\left(\frac{a+1}{2}\right)}{2^{\frac{d-a+1}{2}} \pi^{\frac{d}{2}}}$$

Using the fact that $\Gamma(x)$ is monotonically increasing $\forall x > 1$:

$$\epsilon = \frac{v_d \cdot C_\alpha}{\sigma_{min}^{\alpha/2}} \int_0^\infty k(t) t^{d+\alpha} dt = \frac{\pi^{d/2}}{\Gamma(1+d/2)} \cdot C_\alpha \cdot \frac{\Gamma\left(\frac{d+\alpha+1}{2}\right)}{2^{\frac{1-\alpha}{2}} \pi^{\frac{d}{2}}}$$

$$= \frac{C_\alpha}{\Gamma(1+d/2) \cdot 2^{\frac{1-\alpha}{2}}} \cdot \Gamma\left(\frac{d+\alpha+1}{2}\right)$$

$$\leq \frac{C_\alpha}{\Gamma(1+d/2) \cdot 2^{\frac{1-\alpha}{2}}} \cdot \Gamma\left(\frac{d+2}{2}\right) = C_\alpha 2^{\frac{\alpha-1}{2}}$$

Notice that in our case, since the function is $\alpha$-Hölder continuous and its support size is $|\Theta|$:

$$f_{max} - f_{min} \leq C_\alpha \Delta_{max}^\alpha(\Theta) \Rightarrow \|f\|_\infty \leq C_\alpha \Delta_{max}^\alpha(\Theta) + \frac{1}{|\Theta|}$$

Where $\Delta_{max}(\Theta)$ is the maximum $L_1$ distance between two parameters in $\Theta$.

Using the fact that $\sqrt{\Gamma(2x)} > \Gamma(x)$:

$$C_\infty = 8d\sqrt{v_d \cdot \|f\|_\infty}\left(\int_0^\infty k(t) \cdot t^{d/2} dt + 1\right) + 64d^2 \cdot k(0)$$

$$= 8d\sqrt{\frac{\pi^{d/2}}{\Gamma(1+d/2)} \cdot \|f\|_\infty}\left(\frac{\Gamma\left(\frac{d+2}{4}\right)}{2^{\frac{d+2}{4}} \pi^{\frac{d}{2}}} + 1\right) + \frac{64d^2}{\sqrt{(2\pi)^d}}$$

$$\leq 16d\sqrt{\pi^{d/2} \cdot \|f\|_\infty} 2^{\frac{-d-2}{4}} \pi^{\frac{-d}{2}} + \frac{64d^2}{\sqrt{(2\pi)^d}} \leq \frac{16d\sqrt{C_\alpha \Delta_{max}^\alpha(\Theta) + \frac{1}{|\Theta|}}}{\sqrt{2}(2\pi)^{\frac{d}{4}}} + \frac{64d^2}{(2\pi)^{\frac{d}{2}}}$$

Concluding:

$$\sup_{x\in\mathbb{R}^d}\left|\widehat{f_{\mathbf{H}}}(x) - f(x)\right| < (\epsilon + C_\infty) \cdot \left(\frac{\log n}{n}\right)^{\frac{\alpha}{2\alpha+d}}$$

$$\leq \left(C_\alpha 2^{\frac{\alpha-1}{2}} + \frac{16d\sqrt{C_\alpha \Delta_{max}^\alpha(\Theta) + \frac{1}{|\Theta|}}}{\sqrt{2}(2\pi)^{\frac{d}{4}}} + \frac{64d^2}{(2\pi)^{\frac{d}{2}}}\right) \cdot \left(\frac{\log n}{n}\right)^{\frac{\alpha}{2\alpha+d}}$$

$\square$

## A.5 Bounds for a Truncated Estimator

**Remark 2**. The result of Theorem 6 also holds when truncating the KDE estimate to a support $\Theta$.

*Proof of Remark 2.* Let $f_1 \in \mathcal{P}(\Theta)$ be a PDF where $\Theta$ is of finite size $|\Theta|$ and dimension $d$ and $f_2 \in \mathcal{P}(\mathbb{R}^d)$ another PDF such that $\|f_1 - f_2\|_\infty \le U$ (where $U < 1$). So:

$$\|f_1 - f_2^T\|_\infty \le \frac{(|\Theta| + 1) \cdot U}{1 - |\Theta| U}$$

Where $f_2^T$ is the truncated version of $f_2$ ($f_2^T(\theta) = \frac{f_2(\theta)}{\int_{\theta \in \Theta} f_2(\theta) d\theta}$ for $\theta \in \Theta$, else 0) Let $r = \int_{\theta \in \Theta} f_2(\theta) d\theta$, we can bound $1 - r$:

$$1 - r = \int_{\theta \in \Theta} (f_1(\theta) - f_2(\theta)) \, d\theta \le \int_{\theta \in \Theta} |f_1(z) - f_2(z)| \, dz \le |\Theta| \cdot \|f_1 - f_2\|_\infty$$

So:

$$\|f_1 - f_2^T\|_\infty \le \left\| f_1 - \frac{f_2}{r} \right\|_\infty = \frac{1}{r} \cdot \|f_2 - r \cdot f_1\|_\infty = \frac{1}{r} \cdot \|f_2 - f1 + (1 - r) \cdot f_1\|_\infty$$

$$\le \frac{1}{r} \left( \|f_2 - f_1\|_\infty + \|(1 - r) \cdot f_1\|_\infty \right)$$

Concluding:

$$\|f_1 - f_2^T\|_\infty \le \frac{1}{1 - |\Theta| U} (U + |\Theta| \cdot U) = \frac{(1 + |\Theta|) \cdot U}{1 - |\Theta| U}$$

$\square$

## A.6 Bounds for Discrete and Finite Parametric Space

**Remark 3**. In the case of a discrete and finite parametric space there is no need for the KDE, but we can still use our model based approach, and estimate the prior using the empirical distribution $\widehat{P}_{emp}(M) = \widehat{n}(M)/n$, $\forall M \in \mathcal{M}$, where $\widehat{n}(M)$ is the number of occurrences of $M$ in the training set. We can bound the $L_1$ error of this estimator using the Bretagnolle-Huber-Carol inequality [35], and achieve that with probability at least $1 - 1/n^\alpha$ we have that, $\mathcal{R}_T(\pi^*_{\widehat{P}_{emp}}) \le 2 C_{max} T \sqrt{2 \left( \alpha \log (n \log 2) + |\mathcal{M}| + 1 \right) / n}$.

*Proof of Remark 3.*

$$\Pr \left[ \sum_{i=1}^{|\mathcal{M}|} \left| \frac{\widehat{n}_i}{n} - p_i \right| \ge \lambda \right] \le 2^{|\mathcal{M}|+1} e^{-n\lambda^2/2}$$

$$2^{|\mathcal{M}|+1} e^{-n\lambda^2/2} = n^{-\alpha}$$

$$\frac{1}{\log (2)} e^{|\mathcal{M}|+1-\frac{n\lambda^2}{2}} = n^{-\alpha}$$

$$|\mathcal{M}| + 1 - n\lambda^2/2 = \log \left( \log (2) n^{-\alpha} \right)$$

$$n\lambda^2/2 = \alpha \log (\log (2) n) + |\mathcal{M}| + 1$$

$$\lambda = \sqrt{(\alpha \log (n \log 2) + |\mathcal{M}| + 1) \frac{2}{n}}$$

So with probability at least 1-1/n we have:

$$\sum_{i=1}^{|\mathcal{M}|} \left| \frac{\widehat{n}_i}{n} - p_i \right| \le \sqrt{(\alpha \log (n \log 2) + |\mathcal{M}| + 1) \frac{2}{n}}$$

By using Lemma 3 we get the result.

$\square$

## A.7 The History Dependent Simulation Lemma

**Lemma 7.** For any history-dependent policy $\pi$ and any parametric mapping $g$ that satisfies Assumption 5, the following holds for any $\theta_1, \theta_2 \in \Theta$:

$$\left| L_{M=g(\theta_1),\pi} - L_{M=g(\theta_2),\pi} \right| \leq C_{max} C_g \left\| \theta_1 - \theta_2 \right\|_1 \cdot T^2.$$

*Proof of Lemma 7.* For ease of notation, our proof is for the case of discrete state and action spaces and discrete range of the cost function. Yet, this proof can easily be extended to the more general continuous case by replacing the sums with integrals.

The history at step t:

$$h_t = \{s_0, a_0, c_0, s_1, a_1, c_1 \ldots, s_t\}$$

The cost distribution at step t for a deterministic, history-dependent policy and mdp $M$:

$$C_M^\pi(h_t) := C_M(c_t \mid s_t, \pi(h_t))$$

And the average cost:

$$\bar{C}_M^\pi(h_t) := \sum_{c_t} c_t \cdot C_M(c_t \mid s_t, \pi(h_t))$$

The value function:

$$V_{t,M}^\pi(h_t) = \mathbb{E}_{\pi,M} \left[ \sum_{t'=t}^{T} C_M^\pi(h_{t'}) \mid h_t \right]$$

The RL loss:

$$L_{M,\pi} = \mu^T V_{0,M}^\pi = \mathbb{E}_{\pi,M} \left[ \sum_{t=0}^{T-1} C_M^\pi(h_t) \right]$$

Where $\mu$ is the initial state distribution.

For $t = T - 1$:

$$V_{T-1,M}^\pi(h_{T-1}) = \bar{C}_M^\pi(h_{T-1})$$

For $t < T - 1$:

$$V_{t,M}^\pi(h_t) = \bar{C}_M^\pi(h_t) + \sum_{c_t,s_{t+1}} P_M(c_t, s_{t+1} \mid s_t, \pi(h_t)) V_{t+1,M}^\pi(\{h_t, \pi(h_t), c_t, s_{t+1}\})$$

For two MDPs $M$ and $M'$ such that $\forall s \in \mathcal{S}$ and $\forall a \in \mathcal{A}$:

$$\sum_{c,s'} \left| P_M(c, s' \mid s, a) - P_{M'}(c, s' \mid s, a) \right| = \epsilon$$

We have:

$$\bar{C}_M^\pi(h_t) - \bar{C}_{M'}^\pi(h_t) = \sum_{c_t} c_t \cdot (P_M(c_t \mid s_t, \pi(h_t)) - P_{M'}(c_t \mid s_t, \pi(h_t)))$$

$$\left| \bar{C}_M^\pi(h_t) - \bar{C}_{M'}^\pi(h_t) \right| \leq C_{max} \cdot \sum_{c_t} \left| P_M(c_t \mid s_t, \pi(h_t)) - P_{M'}(c_t \mid s_t, \pi(h_t)) \right| \leq C_{max}\epsilon$$

For ease of notation we define $h_{t+1} = \{h_t, \pi(h_t), c_t, s_{t+1}\}$:

$$V_{t,M}^\pi(h_t) - V_{t,M'}^\pi(h_t) = \bar{C}_M^\pi(h_t) + \sum_{c_t,s_{t+1}} P_M(c_t, s_{t+1} \mid s_t, \pi(h_t)) V_{t+1,M}^\pi(h_{t+1})$$

$$- \bar{C}_{M'}^\pi(h_t) - \sum_{c_t,s_{t+1}} P_{M'}(c_t, s_{t+1} \mid s_t, \pi(h_t)) V_{t+1,M'}^\pi(h_{t+1})$$

$$\left| V_{t,M}^{\pi}(h_t) - V_{t,M'}^{\pi}(h_t) \right| \leq \left| C_M^{\pi}(h_t) - C_{M'}^{\pi}(h_t) \right| +$$
$$+ \sum_{c_t, s_{t+1}} \left| P_M(c_t, s_{t+1} \mid s_t, \pi(h_t)) V_{t+1,M}^{\pi}(h_{t+1}) \right.$$
$$- P_M(c_t, s_{t+1} \mid s_t, \pi(h_t)) V_{t+1,M'}^{\pi}(h_{t+1})$$
$$+ P_M(c_t, s_{t+1} \mid s_t, \pi(h_t)) V_{t+1,M'}^{\pi}(h_{t+1})$$
$$\left. - P_{M'}(c_t, s_{t+1} \mid s_t, \pi(h_t)) V_{t+1,M'}^{\pi}(h_{t+1}) \right|$$

So:

$$\left| V_{t,M}^{\pi}(h_t) - V_{t,M'}^{\pi}(h_t) \right| \leq C_{max}\epsilon +$$
$$+ \sum_{c_t, s_{t+1}} P_M(c_t, s_{t+1} \mid s_t, \pi(h_t)) \cdot \left| V_{t+1,M}^{\pi}(h_{t+1}) - V_{t+1,M'}^{\pi}(h_{t+1}) \right|$$
$$+ V_{t+1,M'}^{\pi}(h_{t+1}) \cdot \left| P_M(c_t, s_{t+1} \mid s_t, \pi(h_t)) - P_{M'}(c_t, s_{t+1} \mid s_t, \pi(h_t)) \right|$$

We know that $V_{t+1,M}^{\pi}(h_{t+1}) \leq (T-1) C_{max}$, so:

$$\left| V_{t,M}^{\pi}(h_t) - V_{t,M'}^{\pi}(h_t) \right| \leq C_{max}\epsilon + \left\| V_{t+1,M}^{\pi} - V_{t+1,M'}^{\pi} \right\|_{\infty} + (T-1) C_{max}\epsilon$$
$$= TC_{max}\epsilon + \left\| V_{t+1,M}^{\pi} - V_{t+1,M'}^{\pi} \right\|_{\infty}$$

Applying the above rule recurrently from $t = 0$ to $t = T - 2$:

$$\left| V_{0,M}^{\pi}(h_0) - V_{0,M'}^{\pi}(h_0) \right| \leq C_{max}\epsilon T^2$$

Plugging the result for the loss difference:

$$|L_{M,\pi} - L_{M',\pi}| = \mu^T \left| V_{0,M}^{\pi} - V_{0,M'}^{\pi} \right| \leq C_{max}\epsilon T^2$$

We receive the result by using assumption 5. $\qquad\square$

## A.8 PCA Error Bounds for Parametric Spaces with Low Dimensional Structure

**Lemma 8**. Under Assumptions 4 and 6, we have that:

$$\mathbb{E}\left[ R\left( \widehat{P}_{d'} \right) \right] \leq \min\left( \frac{8C_{sg}^2 \sqrt{d'} tr(\Sigma)}{\sqrt{n}}, \frac{64C_{sg}^4 tr^2(\Sigma)}{n\left( \lambda_{d'} - \lambda_{d'+1} \right)} \right) + \epsilon \cdot (d - d').$$

*Proof of Lemma 8.* A well known property of the PCA [27]:

$$\min_{P \in \mathcal{P}_{d'}} R(P) = \sum_{i=d'+1}^{d} \lambda_i$$

So:

$$\mathbb{E} R\left( \widehat{P}_{d'} \right) \leq \left( \frac{8C_{sg}^2 \sqrt{d'} tr(\Sigma)}{\sqrt{n}}, \frac{64C_{sg}^4 tr^2(\Sigma)}{n\left( \lambda_{d'} - \lambda_{d'+1} \right)} \right) + \sum_{i=d'+1}^{d} \lambda_i$$
$$\leq \left( \frac{8C_{sg}^2 \sqrt{d'} tr(\Sigma)}{\sqrt{n}}, \frac{64C_{sg}^4 tr^2(\Sigma)}{n\left( \lambda_{d'} - \lambda_{d'+1} \right)} \right) + \epsilon \cdot (d - d')$$

Where we used Theorem 2 for the first inequality and assumption 6 for the second.

$\qquad\square$

## A.9 Regret Bounds by Dimensionality Reduction

**Theorem 9**. Let $\widehat{f}_G^{d'}$ be the approximation of $f$ as defined in Section 4.1. Under Assumptions 1, 4, 6, and 7 we have with probability at least $1 - 1/n$:

$$\mathcal{R}_T\left(\pi_{\widehat{f}_G^{d'}}^*\right) \leq 2C_{max}T\left|\Theta_L\right|C_{d'}\left(\frac{\log n}{n}\right)^{\frac{\alpha'}{2\alpha'+d'}} + 2C_{max}T^2C_g\sqrt{\min\left(\frac{C'_{sg}\sqrt{d'}}{\sqrt{n}}, \frac{C'^2_{sg}}{n\Delta_{\lambda,d'}}\right)} + \epsilon(d - d'),$$

where $C_{d'} = C_{\alpha'}2^{\frac{\alpha'-1}{2}} + \dfrac{16d'\sqrt{C_{\alpha'}\Delta_{max}^{\alpha'}(\Theta_L)+\frac{1}{|\Theta_L|}}}{\sqrt{2}(2\pi)^{\frac{d'}{4}}} + \dfrac{64d'^2}{(2\pi)^{\frac{d'}{2}}}$, $C'_{sg} = 8C^2_{sg}tr(\Sigma)$ and $\Delta_{\lambda,d'} = \lambda_{d'} - \lambda_{d'+1}$

*Proof of Theorem 9.*

$$\mathcal{L}_{f_\theta}(\pi) = \mathbb{E}_{\theta\sim f_\theta}L_{\theta,\pi} = \int L_{\theta,\pi}f_\theta(\theta)d\theta = \int \left(L_{\theta,\pi} - L_{\widehat{P}_{d'}\cdot\theta,\pi} + L_{\widehat{P}_{d'}\cdot\theta,\pi}\right)f_\theta(\theta)d\theta$$

$$= \int \left(L_{\theta,\pi} - L_{\widehat{P}_{d'}\cdot\theta,\pi}\right)f_\theta(\theta)d\theta + \int L_{\widehat{P}_{d'}\cdot\theta,\pi}f_\theta(\theta)d\theta$$

$$= \int \left(L_{\theta,\pi} - L_{\widehat{P}_{d'}\cdot\theta,\pi}\right)f_\theta(\theta)d\theta + \int L_{\widehat{P}_{d'}\cdot\theta,\pi}f_\theta(\theta)d\theta$$

$$\overset{(*)}{=} \int \left(L_{\theta,\pi} - L_{\widehat{P}_{d'}\cdot\theta,\pi}\right)f_\theta(\theta)d\theta + \int_{\theta_L\in\Theta_L}\left(\int_{\theta_L^\perp\in\Theta_L^\perp(\theta_L)} L_{\widehat{P}_{d'}\cdot\theta_L^\perp,\pi}f_\theta(\theta_L^\perp)d\theta_L^\perp\right)d\theta_L$$

$$\overset{(**)}{=} \int \left(L_{\theta,\pi} - L_{\widehat{P}_{d'}\cdot\theta,\pi}\right)f_\theta(\theta)d\theta + \int_{\theta_L\in\Theta_L}\left(L_{W_L^T\cdot\theta_L,\pi}\int_{\theta_L^\perp\in\Theta_L^\perp(\theta_L)} f_\theta(\theta_L^\perp)d\theta_L^\perp\right)d\theta_L$$

$$= \int \left(L_{\theta,\pi} - L_{\widehat{P}_{d'}\cdot\theta,\pi}\right)f_\theta(\theta)d\theta + \int L_{W_L^T\cdot\theta_L,\pi}f_{\theta_L}(\theta_L)d\theta_L$$

$(*)$ Fubini theorem holds since $\int L_{\widehat{P}_{d'}\cdot\theta,\pi}f_\theta(\theta)d\theta \leq \infty$

$(**)$ Since $W_L^T\cdot\theta_L = \widehat{P}_{d'}\cdot\theta_L^\perp, \forall\theta_L^\perp\in\Theta_L^\perp(\theta_L)$.

And we know that:

$$\mathcal{L}_{\widehat{f}_G^{d'}}(\pi) = \mathbb{E}_{\theta\sim\widehat{f}_G^{d'}}L_{\theta,\pi} = \int L_{W_L^T\cdot\theta_L,\pi}\widehat{f}_G(\theta_L)d\theta_L$$

Subtracting the two and taking the absolute value and using the triangle inequality:

$$\left|\mathcal{L}_{f_\theta}(\pi) - \mathcal{L}_{\widehat{f}_G^{d'}}(\pi)\right| \leq \int\left|L_{\theta,\pi} - L_{\widehat{P}_{d'}\cdot\theta,\pi}\right|f_\theta(\theta)d\theta + \int L_{W_L^T\cdot\theta_L,\pi}\left|f_{\theta_L}(\theta_L) - \widehat{f}_G(\theta_L)\right|d\theta_L$$

Starting with the first term:

$$\int\left|L_{\theta,\pi} - L_{\widehat{P}_{d'}\cdot\theta,\pi}\right|f_\theta(\theta)d\theta \overset{(*)}{=} C_{max}C_gT^2\cdot\int\left|\theta - \widehat{P}_{d'}\cdot\theta\right|f_\theta(\theta)d\theta$$

$$\overset{(**)}{=} C_{max}C_gT^2\cdot\sqrt{\int\left(\theta - \widehat{P}_{d'}\cdot\theta\right)^2 f_\theta(\theta)d\theta}$$

$$= C_{max}C_gT^2\cdot\sqrt{\mathbb{E}R\left(\widehat{P}_{d'}\right)}$$

$$\overset{(***)}{\leq} \sqrt{\min\left(\frac{8C^2_{sg}\sqrt{d'}tr(\Sigma)}{\sqrt{n}}, \frac{64C^4_{sg}tr^2(\Sigma)}{n\left(\lambda_{d'} - \lambda_{d'+1}\right)}\right) + \epsilon\cdot(d - d')}$$

$(*)$ Using Lemma 7

$(\ast\ast)$ Using the fact that $\sqrt{x}$ is concave, and using Jensen inequality ($\mathbb{E}[\sqrt{x}] \leq \sqrt{\mathbb{E}[x]}$):

$(\ast\ast\ast)$ Using Lemma 8

The second term can be bounded using lemma 5:

$$\int L_{W_L^T \cdot \theta_L, \pi} \left| f_{\theta_L}(\theta_L) - \widehat{f}_G(\theta_L) \right| d\theta_L \leq |\Theta_L| \, C_{d'} C_{max} T \left( \frac{\log n}{n} \right)^{\frac{\alpha}{2\alpha + d'}}$$

Adding the two terms and using the same argument as in the proof from Section A.2 gives the result. $\qquad\square$

## A.10 VariBad Dream Implementation Details

In our implementation (which is formulated in Algorithm 1), we first run the regular VariBad training scheme for $I_W$ warm-up iterations (because the dream environments at the very start of the training are uninformative). After the warm-up period, at each iteration we insert the last encoded latent vector from each of the real environments (i.e after $H$ steps) into a latent pool. Every $I_{KDE}$ iterations we updated the KDE estimation. At each iteration we sample $n_{dream}$ vectors from the KDE and pass one to each dream environment worker. Each dream environment will use this latent vector and the reward decoder to assign rewards.

---

**Algorithm 1** VariBad Dream

---

**Require:** $\{M_i\}_{i=1}^N \in \mathcal{M}^N$ The training MDPs,
$\qquad\qquad R, D \in \mathbb{N}$ Number of real and dream agents respectively
$\qquad\qquad I_W, I_T \in \mathbb{N}$ Number of warmup and training iterations respectively
$\qquad\qquad I_{KDE} \in \mathbb{N}$ KDE update interval
$\quad real\_workers \leftarrow \{real\_worker()\}_{i=1}^R$
$\quad dream\_workers \leftarrow \{dream\_worker()\}_{i=1}^D$
$\quad latents\_pool \leftarrow \{\}$
$\quad$**for** $i \leftarrow 0$ *to* $I_W$ **do**
$\qquad latents\_pool \leftarrow latents\_pool \cup real\_workers.run\_episode()$ $\qquad\qquad$ ▷ Warmup iterations
$\quad$**end for**
$\quad$**for** $i \leftarrow 0$ *to* $I_T$ **do**
$\qquad$**if** $i \mod I_{KDE} = 0$ **then**
$\qquad\qquad dream\_workers.kde \leftarrow kde(latents\_pool)$
$\qquad\qquad latents\_pool \leftarrow \{\}$
$\qquad$**end if**
$\qquad latents\_pool \leftarrow latents\_pool \cup real\_workers.run\_episode()$ $\qquad\qquad$ ▷ Main iterations
$\qquad dream\_workers.run\_episode()$
$\qquad VariBad.vae\_update()$ $\qquad\qquad\qquad\qquad\qquad$ ▷ Original VAE update
$\qquad VariBad.policy\_update()$ $\qquad\qquad\qquad\qquad\qquad$ ▷ Original policy update
$\quad$**end for**
$\quad$**function** *real\_worker.run\_episode*
$\qquad posterior\_latents \leftarrow \{\}$
$\qquad$**for** *real\_worker in real\_workers* **do**
$\qquad\qquad real\_worker.mdp = random\_sample(\{M_i\}_{i=1}^N)$
$\qquad\qquad VariBad\_rollout(real\_worker.mdp)$ $\qquad$ ▷ Steps in the environment and buffers updates
$\qquad\qquad posterior\_latents \leftarrow posterior\_latents \cup vae.posterior$
$\qquad$**end for**
$\qquad$*return posterior\_latents*
$\quad$**end function**
$\quad$**function** *dream\_worker.run\_episode*
$\qquad$**for** *dream\_worker in dream\_workers* **do**
$\qquad\qquad dream\_worker.latent = dream\_workers.kde.sample()$
$\qquad\qquad curr\_mdp \leftarrow vae.decoder(dream\_worker.latent)$
$\qquad\qquad\qquad\qquad\qquad$ ▷ MDP's transitions and reward defined by the decoder's outputs
$\qquad\qquad VariBad\_rollout(curr\_mdp)$ $\qquad\qquad\qquad\qquad$ ▷ Original episode rollout
$\qquad$**end for**
$\quad$**end function**

---

Our implementation is based on the open-source code of Zintgraf et al [39], which can be found in `https://github.com/lmzintgraf/varibad`.

The code implementing VariBad Dream, and the details on how to reproduce all the experiments presented in this paper, can be found in `https://github.com/zoharri/MBRL2`.

Hyperparameters for VariBad:

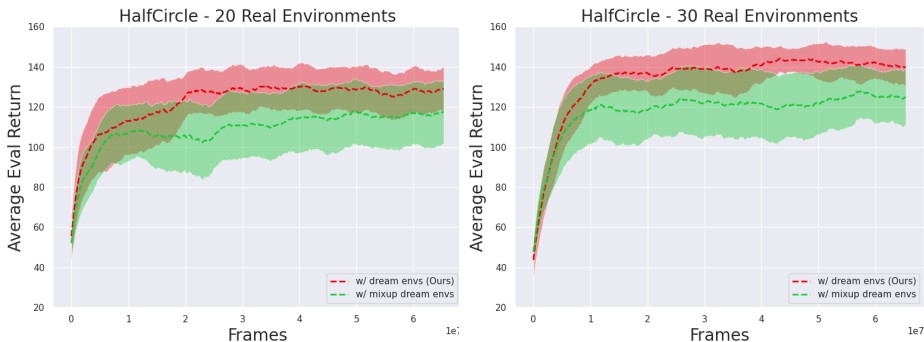

Figure 4: Average return on HalfCircle with KDE and mixup dream environments. The average is shown in dashed lines, with the 95% confidence intervals (15 random seeds).

| Rollout horizon | 100 |
|---|---|
| Number of rollouts | 2 |
| RL algorithm | PPO |
| Epochs | 2 |
| Minibatches | 4 |
| Max grad norm | 0.5 |
| Clip parameter | 0.05 |
| Value loss coeff. | 0.5 |
| Entropy coeff. | 0.01 |
| Gamma | 0.97 |
| Weight of KL term in ELBO | 1 |
| Policy learning rate | 7e-4 |
| VAE learning rate | 1e-3 |
| VAE batch size | 5 |
| Task embedding size | 5 |
| Policy architecture | 2 hidden layers, 128-dim each, TanH activations |
| Encoder architecture | States/actions/rewards encoder: FC layer 32/16/16 dim, GRU with hidden size 128 , output layer of dim 5, ReLu activations |
| Reward decoder architecture | 2 hidden layers, 64 and 32 dims, ReLu activations |
| Reward decoder loss function | Mean squared error |

The Hyper parameters for VariBad Dream are the same as for VariBad with the following additional parameters:

| Number of warm-up iterations | 5000 |
|---|---|
| KDE update interval | 3 |

For the case of 20/30 sample sizes we chose 4/6 dream environment workers and 12/10 real environment workers. The intuition is that as we have more training environments, the better the latent representation and KDE are, and we can rely more on the dream environments.

Similarly to [39] and [21], we used only the reward decoder (and not the state decoder) due to better empirical results.

## A.11 KDE and Mixup Dream Environments Comparison

In this section we compare between using the proposed KDE and the simple Mixup approach to generate dream environments. While Mixup approaches usually aim at solving out of distribution generalization, we use it here as a baseline for the prior estimation. We emphasize that we don't claim to solve OOD generalization. In the Mixup approach, given $N$ latent vectors of real environments, we sample $N$ coefficients from the Dirichlet distribution $\vec{\alpha} = Dir(1, \ldots, 1)$ and use them to calculate a weighted average for the dream environment.

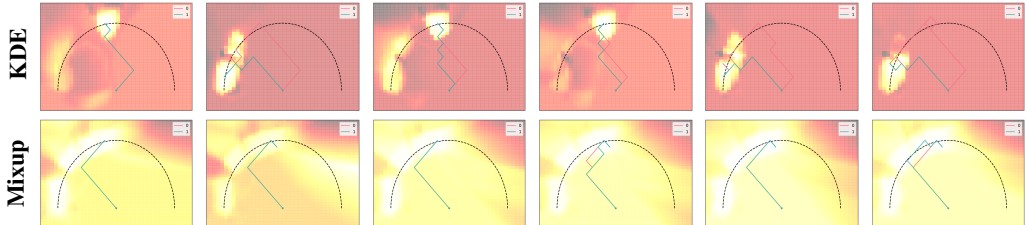

Figure 5: Comparison of dream environments' reward maps, generated during the training using 20 real environments. On the top row - our suggested KDE method. On the bottom row - the Mixup method. Each column corresponds to a different training iteration with a 1000 iteration interval between each one. The trajectory of the policy for the sampled dream environment is plotted on top of the reward map as well.

In Figure 4 we compare the average evaluation return of the proposed KDE dream environments to the Mixup dream environments. Evidently, our approach achieved higher return, both for $N_{train} = 20$ and $N_{train} = 30$.

To further analyze the differences between the two approaches and to better understand the dream environments, we visualize the reward map of the dream environments, as can be seen in Figure 5. In order to visualize the reward map for a given latent vector (sampled using either KDE or Mixup), we pass a discrete grid of states and the latent vector to the reward decoder and draw a heatmap of the results. In Figure 5 we plot multiple sampled latent vectors, which were observed during the training. We can see that the KDE produces much more realistic and variable dream environments than the Mixup approach, explaining its superior performances.

## A.12  Ant Goal Environment

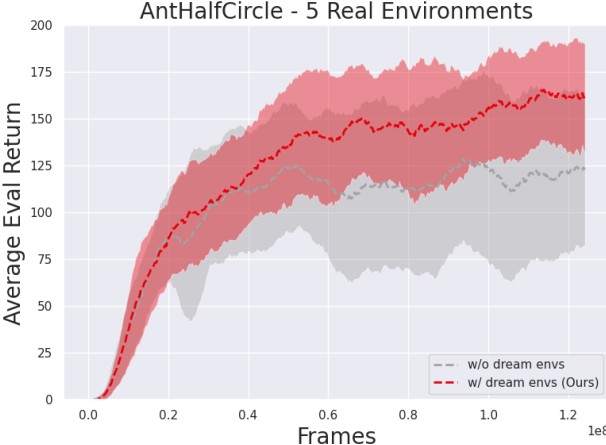

Figure 6: Average return on the Ant Goal environment with and without dream environments. The average is shown in dashed lines, with the 95% confidence interval (8 random seeds).

Our PAC bounds showed that the determining factor in generalization is not the dimensionality of the MDP (i.e., the dimension of the state and action spaces), but the dimensionality of the underlying MDP parameter space $\Theta$.

In this section, we demonstrate this claim by running VariBad Dream on a high-dimensional continuous control problem – a variant of the HalfCircle environment (Figure 1) with an Ant robot agent, simulated in MuJoCo [34]. Note that while the space $\Theta$ is low dimensional, similarly to the point robot HalfCircle experiments, each MDP has a high dimensional state and action space.

In order to make the baseline VariBAD method work on this environment properly, we increased the goal size from 0.2 (which was used in the HalfCircle environment) to 0.3. We found that this change was necessary for VariBAD to explore effectively and reach the goal during training.

In Figure 6 we compare VariBAD and VariBAD Dream on this environment. Similarly to the point robot results, we observe an advantage for VariBAD Dream when the number of training MDPs is small ($N_{train} = 5$). For $N_{train} = 10$ we did not observe significant advantage for VariBAD Dream, as, similarly to the $N_{train} = 40$ case in point robot, the training samples already cover the space $\Theta$ adequately in most runs.

### A.13 Compute Specification

We ran the experiments on a Standard_NC24s_v3 Azure machine consisting of 4 NVIDIA Tesla V100 GPUs. Running 15 seeds at a time took a total of $\sim$24 hours.