# OpenReview forum: "Meta Reinforcement Learning with Finite Training Tasks - a Density Estimation Approach "
_NeurIPS.cc/2022/Conference — NeurIPS 2022 Accept_

### Official Review · Reviewer_RVe9 · 2022-07-05

**Rating:** 6
**Confidence:** 5
**Soundness:** 3 good
**Presentation:** 2 fair
**Contribution:** 3 good

**Summary:**

This work proposes an imaginary task generation method for meta-RL tasks with a limited number of training tasks using the prior estimated by kernel density estimation (KDE). The authors provide a theoretical bound for the regret of the estimation and empirically evaluate the method in conjunction with variBAD [1] on the HalfCircle environment.

[1] Zintgraf et al., Varibad: A very good method for bayes-adaptive deep RL via meta-learning.

**Questions:**

1. Is there any additional methods the authors tried to improve the generalization of the generated tasks? Because it seems that the policy may overfit the KDE-estimated prior. E.g., generating contexts with perturbation on the prior.

2. It would be great if the authors provide a table of all notations used in this paper in Appendix, as some of the notations are confusing and similar.

3. It would be great if the authors review the reference and update them to the recent conference proceedings.
e.g., [18] arXiv->ICLR2019, [19] CorR->NeurIPS 2021, [20] arXiv->NeurIPS 2020...



**Limitations:**

This work is based on in-distribution parametric meta-RL with finite training tasks.
Although it is not the focus of this work in the current state, it is hard to generalize to non-parametric, non-stationary, and out-of-distribution meta-RL tasks [3].

[3] Yu et al., Meta-World: A Benchmark and Evaluation for Multi-Task and Meta Reinforcement Learning

**Strengths And Weaknesses:**

***Strengths***

Overall, I think the strength of this study lies in its novelty and theoretical analysis.

1. Novelty

There are some previous works that generate imaginary tasks using the latent model as LDM [2]. However, this is the first work that directly estimates the density using KDE and uses the density to sample new tasks. This novel idea sounds reasonable and effective for parametric meta-RL tasks with finite tasks.

2. Solid theoretical bound

I appreciate the authors for the solid derivations for the regret bounds in section 2.2. Although the conclusion that the bound is proportional to the dimension of the parameter space is somewhat natural, it has a great meaning that it was shown through a solid proof. Especially the use of PCA to reduce dimension is well supported by this theory.

***Weaknesses***

On the other hand, I think there are weaknesses in practical application and experimentation. I think the contribution of this work can be more significant when the empirical evaluation and analysis are conducted as follows.

1. Ambiguous link between theory and algorithm

I felt the transition from section 4 to section 6 to be somewhat abrupt. Maybe providing a pseudo-code of variBAD+KDE would prevent confusion. It is unclear whether the proposed method would work on the latent space generated by the encoder of VAE, which may deviate from the assumptions made.

I know that the VAE of variBAD [1] also does some implicit density estimation based on multivariate Gaussian distribution initialized to standard normal distribution. Due to the nature of the VAE, the latent space generated by variBAD may already be regular (continuous and complete). Maybe applying the KDE on top of this latent space makes the latent space more regular?  One possibile way to show this is an empirical comparison on the latent space for unseen tasks as the t-SNE plot in [2], Figure 9.

Also, I'm curious how regular and general the generated tasks will be given unseen context or states without regularizations as a dropout in [2]. I appreciate the visualization in Figure 4, but I'm not sure if the reward decoder would generate high rewards for states that were not rewarding during training for any training tasks. Maybe 20~30 tasks are enough to contain all states in a light blue circle in Figure 1?

As the authors claim, it could have been because of the special geometry of the HalfCircle's task distribution. Therefore I consider testing only one environment as a weakness of this work.

2. Limited experiment domains

The HalfCircle environment in this work is a very meaningful and intuitive domain. However, it is limited to a one-dimensional task space, therefore it is hard to conclude that the proposed method would outperform generally. Maybe the proposed method works well by overfitting the policy for the learned task prior. To emphasize the contribution of this work for finite training tasks, I strongly recommend evaluating at least tasks evaluated by variBAD [1] or out-of-distribution tasks evaluated by LDM [2].


[1] Zintgraf et al., Varibad: A very good method for bayes-adaptive deep RL via meta-learning.

[2] Lee et al., Improving Generalization in Meta-RL with Imaginary Tasks from Latent Dynamics Mixture.

---

> ### Author Response · Authors · 2022-08-02
> **Comment to Reviewer RVe9**
>
>
> We thank the reviewer for the insightful review.
>
> **Paper summary:**
>
> We kindly refer the reviewer to the comment to all reviewers above and emphasize - our main contribution is the SOTA theoretical results, and not the algorithm, which only supplements the theory.
>
> **Link Between Theory and Algorithm:**
>
> Our theoretical results, and specifically, the idealized method in lines 260-263, make several assumptions:
>
> (1) Knowing the mapping $g(\theta)$ from parameters to MDPs
>
> (2) Linear dimensionality reduction (PCA)
>
> (3) Optimal solution of the meta RL problem with the estimated prior (line 184)
>
> (4) Optimal KDE kernel
>
> One could argue that such assumptions are not practical, and therefore our insights are not relevant to realistic deep meta RL algorithms.
>
> Our goal in Section 6 is to connect our idealized method (lines 260-263) with a practical algorithm (VariBAD), and show that our theoretical insights hold, even when assumptions (1-4) above are clearly not satisfied.
>
> In particular, we relate the VAE encoder in VariBAD to non-linear dimensionality reduction (per 2), and the VAE decoder to $g(\theta)$ (per 1). The deep RL component of VariBAD approximately solves the meta RL problem with the estimated prior (per 3), and we use off-the-shelf KDE estimation (per 4).
>
> Our experiments, therefore, supplement the theory and demonstrate that not only our idealized method relates to a practical algorithm, but the KDE insight can actually improve performance.
>
> Following the reviewer’s suggestion, we added pseudocode to the appendix (Section A.11).
>
>
> **Out of Distribution Generalization:**
>
> OOD generalization in meta RL is an important problem, and recent methods such as LDM have made important progress on it. However, OOD generalization is *not the problem we study in this work*. In our setting, as formally established in Section 3, test tasks are sampled from the same distribution as the training tasks.
>
> The connection to OOD generalization may have been implicitly implied by our choice of the LDM baseline in our experiments. We apologize, and added a clarification to the text - we do not make any claim for VariBAD Dream outperforming LDM on out of distribution tasks. We chose LDM simply because we could not find a more relevant baseline for adding some regularization to the latent space.
> For the tasks we considered, KDE performed better. However, our evaluation is not claimed to be exhaustive (an exhaustive evaluation would likely fit in a different paper, and actually, we have no reason to assume that KDE will outperform LDM in general). This does not invalidate our main claim - that the regularization implied by KDE can improve performance, and that such improvement is not trivial to obtain (i.e., on the problems we tried, it was not obtained using LDM).
>
> **“Limited experiment domains”:**
>
> We added an experiment on the MuJoCo sparse ant goal environment (in the comment to all reviewers).
>
>
> **VariBad Latent Space:**
>
> Please see the new experiments in Section A.10, based on Reviewer ST9x’s suggestion, where we do not use the VAE encoder, but apply KDE directly on the task parameter space. These experiments, which show a more pronounced advantage for KDE, show that the advantage is not only due to the VAE’s latent space structure, but mostly due to the KDE.
>
> **“Is there any additional methods the authors tried to improve the generalization of the generated tasks? Because it seems that the policy may overfit the KDE-estimated prior”:**
>
> We hope we understood the question correctly. Note that the KDE kernel width is inversely proportional to the number of samples. Thus, as our theory implies, the KDE distribution will converge to the true prior, and the policy will not overfit. For low sample sizes, the KDE distribution has some bias, but it is still better than the empirical distribution, as demonstrated empirically in our new experiments - observe that KDE leads to a clear improvement, even for low sample sizes, which do not cover the full goal distribution.
>
> **Notations Table:**  We will add such a table to the final version of the paper.
>
> **References:** Thank you for pointing out the issue in our references. We have fixed that (updated in the revised paper)

---

> > ### Comment · Reviewer_RVe9 · 2022-08-06
> > **Thank you for the author response**
> >
> > I appreciate the authors for the response and the revision of the paper.
> >
> > Given that the paper is mostly based on theoretical results, most of my concerns about the experimental evaluation are addressed. Especially, I appreciate the additional experiments the authors added during the rebuttal.

---

### Official Review · Reviewer_6rr7 · 2022-07-08

**Rating:** 6
**Confidence:** 2
**Soundness:** 3 good
**Presentation:** 3 good
**Contribution:** 3 good

**Summary:**

In this paper, PAC bounds for meta reinforcement learning with finite training tasks are derived. The derived bounds is applicable to both continuous and discrete state and action spaces. The derived bounds have exponential dependence on the dimensionality of the task space. To address this, the analysis is extended to the case with dimensionality reduction using principal component analysis. To support the theoretical results, the technique to improve the existing meta-RL method was evaluated in a simple environment.

**Questions:**

- In practice, how should we determine the optimal bandwidth of the Gaussian kernel for KDE? How did the authors determined the optimal bandwidth for the experiment?
- I did not clearly understand how practical the problem setting of meta RL with finite training tasks is. I would appreciate it if the authors provide examples in the real world where meta RL with finite training tasks should be considered.

**Limitations:**

I think that the authors clearly described the limitation of their theoretical analysis. For example, it is clearly described that analysis regarding the dimensionality reduction is limited to PCA, which is a linear method.

**Strengths And Weaknesses:**

Strength:
- Novel and practical PAC bounds for meta-RL are derived
- Unlike the bound proposed by Tamar et al. (2021), the derived bound can be applied to both continuous and discrete state action spaces
- In addition to theoretical results, experimental results are reported

Weakness
- The work is largely based on the prior work by Tamar et al. (2021), and one might think that the contribution is incremental
- The experiments seem simple, while I think that the theoretical results are strong enough

---

> ### Author Response · Authors · 2022-08-02
> **Comment to Reviewer 6rr7**
>
> We thank the reviewer for the insightful review.
>
> **Novelty with Respect to Tamar et al. (2021):**
>
> Although both papers tackle a similar problem, there is **absolutely zero overlap** between our approach and Tamar et al. (2021). Tamar et al. (2021) base their approach on algorithmic stability and show that L2 regularized MDPs satisfy stability, yielding PAC bounds. Our approach is based on PAC bounds for KDE and PCA, combined with a Lipschitz smoothness of the value function, and does not involve algorithmic stability. The reviewer may verify that our proofs **do not build on any result** from Tamar et al. (2021).
> Our **completely different** approach to the problem allowed us to derive **significantly better results**, under significantly more relaxed assumptions, as detailed in Section 4 of our paper.
>
> We also remark that Tamar et al. (2021) did not perform any empirical evaluation, and it is not clear whether their theoretical insights are indeed practical. In comparison, we showed that the KDE idea can yield a clear, yet modest, improvement in practical meta RL.
>
>
> **“The experiments seem simple”:**
>
> We added an experiment on the MuJoCo sparse ant goal environment (see comment to all reviewers). We also added experiments in a simplified setting without the VariBAD VAE based on reviewer ST9x’s suggestion, which further strengthens our claims.
>
> **“In practice, how should we determine the optimal bandwidth of the Gaussian kernel for KDE? How did the authors determine the optimal bandwidth for the experiment?”**
>
> The Gaussian kernel bandwidth is chosen based on the number of samples (inversely proportional, as stated in section 4 of our paper). Popular practical methods include Scott’s rule and Silverman’s [2,3], as well as the bandwidth shown in our paper. In our experiments, we used SciPy’s default Scott’s rule implementation but observed that this choice doesn’t have a big impact on performance.
>
> **“I would appreciate it if the authors provide examples in the real world where meta RL with finite training tasks should be considered”.**
>
> In robotics applications, the number of training tasks is limited. This has motivated the study of offline meta-RL [3,4], where data from a finite set of tasks is collected prior to the meta-RL algorithm. In particular, [5] used offline meta RL for real-world robotic insertion.
>
>
> More generally, all deep meta RL algorithms that we are aware of are trained for a finite number of iterations, and therefore, see a finite number of tasks (even if tasks are randomly generated during training). As mentioned in our response to reviewer ZFq8, an exponential dependence on the task space dimension would render such methods impractical when the dimension is high, as training would simply require too many tasks.
>
>
> [1] Aviv Tamar, Daniel Soudry, and Ev Zisselman. Regularization guarantees generalization in bayesian reinforcement learning through algorithmic stability. Proceedings of the AAAI Conference on Artificial Intelligence, 36(8):8423–8431, 2022
>
> [2] D.W. Scott, “Multivariate Density Estimation: Theory, Practice, and Visualization”, John Wiley & Sons, New York, Chicester, 1992.
>
> [3] B.W. Silverman, “Density Estimation for Statistics and Data Analysis”, Vol. 26, Monographs on Statistics and Applied Probability, Chapman and Hall, London, 1986.
>
> [4] Li et al., Multi-task batch reinforcement learning with metric learning, 2020.
>
> [5] Dorfman et al, Offline Meta Reinforcement Learning -- Identifiability Challenges and Effective Data Collection Strategies, 2021.
>
> [6] Zhao et al., Offline Meta-Reinforcement Learning for Industrial Insertion, 2021.

---

> > ### Comment · Reviewer_6rr7 · 2022-08-04
> > **Definition of "optimal bandwidth"**
> >
> > Thank you for the clarification. I understand that the proposed approach is clearly different from that in Tamar et al. (2021). I still have a question about the bandwidth of KDE. Scott’s rule for the bandwith is just a heuristic method, which is known to work well in practice, but in my undersanding,  it is not theoretically justified to be "optimal." What is the difinition of "optimal bandwidth" in this paper? What condition should be satisfied by the "optimal bandwidth" ?
> >
> > As I'm not the expert on this field, I may have missed some points which are already described in the paper. But I would appreciate it if the authors answer my questions.

---

> > > ### Author Response · Authors · 2022-08-04
> > > **Optimal Bandwidth and Algorithm**
> > >
> > > We defined the optimal bandwidth as the bandwidth that minimizes the bound of Theorem 1, resulting in the bandwidth of Lemma 4 (proof in Section A.3 of the appendix).
> > >
> > > As you can see, our optimal bandwidth and Scott's rule are very similar:
> > > $h_{scott} = n^{-\frac{1}{4+d}}$ and  $h_{ours} = n^{-\frac{1}{2\cdot \alpha+d}}$ (up to a constant, while ignoring log terms), where $\alpha$ is defined in Assumption 3.
> > >
> > > In practice, since we don't know $\alpha$, we used Scott's rule as an approximation to the optimal bandwidth.
> > >
> > > Please let us know if you have further concerns

---

### Official Review · Reviewer_ST9x · 2022-07-11

**Rating:** 6
**Confidence:** 3
**Soundness:** 2 fair
**Presentation:** 2 fair
**Contribution:** 2 fair

**Summary:**

This paper can be separated into two quite different sections.

The theoretical part derives complexity bounds for an algorithm that would have access to the MDP’s parameters (noted theta), that would run a PCA in order to reduce the dimension of that parameter’s space, and learn the density as modeled by an independent gaussian distribution (presented as an example of KDE).
Then, the Bayes-optimal policy can be computed from this learned distribution under better PAC criteria than by using the naive density approximation in the high-dimensional space.

The experimental part considers data augmentation of the training set of MDPs by using the latent space learned by VariBAD. One can link this approach to the theoretical part by comparing the space reduction performed by the VAE of VariBAD with a PCA,
where the high-dimensional parameter is the history itself (through enough episodes to reduce uncertainty), and the low-dimensional projection is the latent state at the end of an episode.


**Questions:**

L 263: why projecting back? It should be sufficient and even easier to train an agent using the reduced representation of priors.

**Limitations:**

Not applicable.

**Strengths And Weaknesses:**

The theoretical part looks correctly driven, and brings up an interesting proven result: it is worth reducing the dimension to concentrate on the low-dimensional manifold that structures the set of MDPs.

However, the experimental part is rather disappointing, as it does not directly verify the first part’s claims, and treats a very different setting (where theta is not observed) with quite a different algorithm, despite the fact that this separated part reveals another interesting result: one can use the VAE’s latent space to augment the data and improve generalisation with less examples. Also, this part is missing higher dimensional MuJoCo environments in order to fairly compare with VariBAD.

In my opinion, this paper could even be divided into two papers that would be worth separated publication:

1) the first including this theoretical part with a good experimental study: where theta is observed, using both discrete and continuous environments (including the half-circle with function approximation),
And comparing to naive baselines (ppo/sac with concatenated state + theta observation, KDE without PCA dimension reduction)

2) the second including the experimental part with a proper method / algorithm description, and adding larger envs (mujoco). Eventually, a theoretical explanation that would rather look at the improvement as the result of a data augmentation.

---

> ### Author Response · Authors · 2022-08-02
> **Comment to reviewer ST9x**
>
> We thank the reviewer for the insightful review.
>
> **Known Parameters Experiments:**
>
> We emphasize that $\theta$ is never known at test time - this is the Meta RL / Bayesian RL setting. Thus, an experiment with “concatenated state + theta observation” doesn’t fall under the investigation in this paper.
>
> What we *do* assume to be known in the theory, yet don’t exploit in VariBAD Dream, is observing the $\theta’s$ of the training MDPs, and knowing $g(\theta)$ - the mapping from $\theta$ to MDP. Motivated by the reviewer’s suggestion, we designed an experiment where this knowledge is exploited. To do this, given the training MDP parameters ($\theta = (x,y)$ positions of the goal), we estimated their density with KDE and used this KDE to sample MDPs in VariBad (exactly as our theoretical paradigm suggests).
> Our results nicely show a much more pronounced advantage to the KDE approach than in VariBAD Dream.
> The average and 95% confidence interval (6 random seeds) on the unseen test environments (of the point robot HalfCircle environment) are as follows (the full details and graphs are in section A.10 of the revised appendix):
>
> |Number of Training Tasks|With KDE|Without KDE|
> |----------------------------------|-------------|-----------------|
> |$5$|$127.6\pm 13.8$|$35.8\pm 16.3$|
> |$10$|$138.8\pm 6.0$|$82.8\pm 21.8$|
> |$20$|$139.3\pm 8.0$|$114.8\pm 19.3$|
> |$30$|$141.8\pm 6.5$|$115.2\pm 26.9$|
>
> Thank you for a great idea!
>
>
> **Higher Dimensional MuJoCo Environments:**
>
> We added an experiment on the MuJoCo sparse ant goal environment (details are in the comment to all reviewers)
>
> **“L 263: why projecting back? It should be sufficient and even easier to train an agent using the reduced representation of priors”:**
>
> We only assume knowledge of $g(\theta)$, where $\theta$ is in the high dimensional space. Thus, we must project back to obtain the MDPs for calculating the estimated Bayes optimal policy (line 184). We clarify this point in the text.
>
>
> **Paper Structure:**
>
> We appreciate the suggestion. However, we believe that with the additional experiments, our presentation demonstrably shows that our theoretical insights hold beyond the limited setting in our proofs (linear PCA, knowing the mapping from theta to MDP, optimal solution of the meta RL problem with the estimated prior).

---

> > ### Comment · Reviewer_ST9x · 2022-08-08
> > **Thanks for the precise response.**
> >
> > The additional experiments and the author's comment satisfied most of my concerns. I agree the structure will benefit and will look more unified with this experiment that is closer to the theoretical context (even though I would have loved to see an experiment involving a set of discrete MDPs perfectly observed, a true PCA etc, just to perfectly illustrate the theory). I raised my score.

---

### Official Review · Reviewer_ZFq8 · 2022-07-12

**Rating:** 6
**Confidence:** 2
**Soundness:** 3 good
**Presentation:** 3 good
**Contribution:** 3 good

**Summary:**

The paper focuses on designing PAC meta RL algorithms. While prior works took a model-free approach of learning regularized history conditioned policies, this work first estimates the underlying task distribution from the training task and then trains a meta RL algorithm on the learned task distribution. The resulting approach obtains state-of-the-art PAC bounds and shows that the approach can be used to improve practical meta RL algorithms (such as VariBAD).

**Questions:**

N/A

**Limitations:**

The paper doesn't explicitly discuss its limitations

**Strengths And Weaknesses:**

**Strengths:**
  1. The paper uses KDE techniques to improve the PAC bounds
  2. The resulting approach is simple which can be used to improve existing meta RL algorithms such as VariBAD

**Weaknesses:**
  1. While the approach obtains better PAC bounds, my concern is that KDE estimation will only work on tasks which have some low-dimensional embedding (such as the reaching task). I don’t think this approach of estimating task distribution and then learning from the task distribution will scale up to more complex settings (say, meta-world ML10 benchmark, Yu et al., 2021, https://arxiv.org/pdf/1910.10897.pdf).

Overall, I do think that the strength outweighs the weakness of the paper. However, I should say that PAC bounds for meta RL isn’t exactly my area of expertise and hence my opinion should be taken with a grain of salt.

---

> ### Author Response · Authors · 2022-08-02
> **Comment to reviewer ZFq8**
>
> We thank the reviewer for the insightful review.
>
> **MetaWorld and High Dimensional Task Distribution:**
>
> We absolutely agree with this statement, but see it as a strength, rather than a weakness of our analysis. In MetaWorld, each individual task is low dimensional (typically 1 variable per task), yet the space of all tasks is high dimensional. Our analysis suggests that pursuing approximately Bayes-optimal policies here would require too many training domains to be practical. This does not mean that there’s nothing to learn - learning transferable features and skills seems like a good idea (yet would require a different theoretical analysis).
> We mention that the recent work of Mandi et al. [1] is in agreement with the observation above. [1] show that for high-dimensional task spaces (e.g., RLBench, similar to MetaWorld), pre-training + fine-tuning outperforms meta-RL methods that adapt at test time (Bayes optimal is a special case of such).
> One contribution of our work is therefore a principled rule of thumb to decide whether to tackle a problem with meta-RL or resort to a less optimal learned exploration (fine tuning can require several orders of magnitude more samples at test time than Bayes optimal exploration), based on the task space dimensionality.
>
> Finally, we mention that the lower-bound example in Proposition 2 of Tamar et al. (2021) can be adapted to our setting (by using the Tabular Mapping), to show that in some cases, an exponential dependence on the dimension cannot be avoided, regardless of the algorithm.
>
> [1] Mandi et al., On the Effectiveness of Fine-tuning Versus
> Meta-reinforcement Learning, 2022. ArXiv 2206.03271.
>
>
> **No Limitation Section:**
>
> We added an explicit summary of limitations to the appendix (Section A.1).

---

### Author Response · Authors · 2022-08-02
**General comments for the AC and all the reviewers**

We thank the reviewers for their efforts and suggestions. Changes in the revised submission are highlighted in blue.

We would like to clarify our contribution.

Our main contribution is the theoretical finite-task analysis of meta RL. Our analysis improves on the theoretical SOTA for this problem, and formally establishes several insights about meta RL, such as the dependence on the task space dimension, and the advantage of regularizing the learned task prior (using KDE). We mention that the reviewers claimed that our theory “has a great meaning” (RVe9), is “Novel and practical” (6rr7), and “brings up an interesting proven result” (ST9x).

The purpose of the experimental part of our paper is to supplement our theory and show that our insights hold beyond the limited setting in our proofs (linear PCA, knowing the mapping from theta to MDP, optimal solution of the meta RL problem with the estimated prior).

Our algorithmic implementation of VariBAD Dream should not be interpreted as the focus of this work, and we do not claim that our method is a SOTA meta RL algorithm. This point should have been clear from our presentation: the empirical section is given very limited attention (1 sentence in the abstract, 1 paragraph in the introduction, and 1 page out of 9 in the paper). We emphasize this point, as all 4 reviews did not contain a single criticism of our theoretical results (Reviewer 6rr7’s claim on limited novelty w.r.t. Tamar et al. is completely wrong - there is simply zero overlap in our approaches), yet only criticized our experiments.

Following the reviewers' suggestions, we added an additional MuJoCo experiment (see below), and an experiment with observable MDP parameters (see reviewer ST9x comment), which we believe significantly strengthens the paper. That said, we kindly ask the reviewers to calibrate their evaluation of our experiments in proportion to our claim - that our theoretical insights are relevant also to practical meta RL. We believe that our empirical investigation, though limited in scope, indeed demonstrates this well.

**Higher Dimensional MuJoCo Environments:**

We ran an experiment on the sparse AntGoal MuJoCo environment. In this domain, which is at least as challenging as the domains in VariBAD (high dimensional control, sparse reward), an approximately Bayes-optimal behavior is easy to validate - the ant must search for the goal along the half-circle.
In order to make the original VariBAD work well on this domain (exploration is challenging with sparse rewards), we increased the goal size from 0.2 (per the HalfCircle environment) to 0.3. In this setting, $N_{train}=5$ goals can cover the goal space, yet on a typical sample, they don’t.
The average and 95% confidence interval (Evaluated with 8 random seeds) on the unseen test environments with our approach: $162.2 \pm 29.7$ and with VariBAD: $123.2 \pm 60.3$. Full details and graphs in section A.13 of the revised appendix.

---

### Meta-Review · Area_Chair_4aYy · 2022-08-24

**Recommendation:** Accept
**Confidence:** Certain

**Metareview:**

The paper provides a theoretical PAC analysis of meta-learning when the task distribution is directly learned. Under the assumption that the task distribution can be approximated in a low-dimensional space, this approach leads to improved bounds wrt to previous literature. The authors further illustrate how the lessons learned in the theory can be translated into a practical algorithm by integrating their kernel density estimation into the VariBAD algorithm.

There is general consensus that the paper makes several interesting contributions. First the authors devise a new algorithmic approach for meta-learning and derive PAC bounds that clearly improve over current results. The results are rigorous and their interpretation is insightful. Furthermore, the authors made a considerable effort in translating their method into an actual algorithm with a non-trivial empirical validation. While more work may be needed to have a full picture of the empirical merits and limitations of the proposed method, I am confident the paper can be built upon by other researchers in the area.

I strongly suggest the authors to integrate their rebuttal into the final version of the paper, in particular the discussion on the dependency on the dimensionality.

**Award:**

No

---

### Decision · Program_Chairs · 2022-09-14

Accept